# Energy Centers in a Smart City as a Platform for the Application of Artificial Intelligence and the Internet of Things

Bohumir Garlik

Department of Building Technical Equipment, Faculty of Civil Engineering Prague, Czech Technical University, 16000 Prague, Czech Republic; bohumir.garlik@fsv.cvut.cz

**Abstract:** A fundamental strategy of addressing the energy performance of buildings (EPB) and creating conditions for energy sustainability in applying renewable energy sources (RES) is the effective management of building technical equipment. The buildings in question are buildings within a cluster of buildings. The building cluster can be considered as a unit that is, in the sense of the system concept, a part of a subdivision of the city and subsequently of the whole urban agglomeration. We are talking about smart cities. The control system of a building and, subsequently, smart cities is a process management system using artificial intelligence (AI). In order to achieve the desired effects, such as "near-zero energy buildings", as required by the European Energy Performance of Buildings Directive (EPBD 3), the application of AI needs to be shifted towards wireless network connection, i.e., "Internet of Things" (IoT) application. The aim is to create conditions for reducing energy consumption, improving environmental comfort in buildings and reducing $CO_2$ emissions. This paper further analyzes the current state of IoT and the implementation thereof in the process management of sustainable energy at the smart city level as a basic element of a smart city system applied to building management.

**Keywords:** EnergyHub; municipal energy center; IoT; artificial intelligence; microgrid; objective function; NZEB





## 1. Introduction

The Internet of Things (IoT) (Internet of Things (IoT). Cisco.com [online]. 2015 [cited on 14 January 2016]. Available at: http://www.cisco.com/c/en/us/solutions/internet-of-things/overview.html, accessed on 9 March 2022) is a recent communication paradigm foreseeing a near future, wherein everyday objects will be equipped with microcontrollers, transceivers for digital communication, and appropriate protocol stacks that will allow them to communicate with each other and with users as well as to become an integral part of the Internet [1]. This used to be called "M2M" (Machine to Machine), then "Connected Devices", "Smart Devices"; however, it is still the same–the Internet of Things (IoT).

The basic building blocks of the Internet of Things consist of three blocks, namely things, network, and cloud. Things are represented by devices that can be connected to the Internet, either via a wired or wireless connection. These things are connected to the cloud by a network used as a communication medium. The cloud is defined as remote servers in data centers where transmitted or retrieved data is stored. This data is detected from things using sensors, e.g., for sensing temperature, position, or ambient air humidity. This information in the form of data is so-called small data because its size is only a few bytes. The next step is to forward small data through the network to the cloud, where it is aggregated and tracked. In this way, larger and larger amounts of data are collected over time, creating so-called big data. According to Cisco, people, machines, and things will generate 850 ZB worth of data by 2021. There is no point in storing such large amounts of data, and only a fraction of the data is useful for storing in a cloud and analyzing later. According to Cisco, approximately 10% are useful, corresponding to 85 ZB [2,3]. Cisco is

also looking at the Internet of Things and estimates that it will consist of 50 billion devices connected to the Internet in the near future (Cisco has also come up with a new term: Internet of Everything. Furthermore, a section called Smart+Connected City (e.g., Traffic, Lightning, Wi-Fi) can be found in the Industry Solutions category. Thus, Cisco is trying to implement various technology solutions for smart cities.

Indeed, this paradigm finds applications in many different fields, such as home automation, industrial automation, medical devices, mobile healthcare, assistive services for the elderly, smart energy management and smart grids, automotive, traffic management, and many others [4]. The usage of IoT is illustrated in Figure 1.

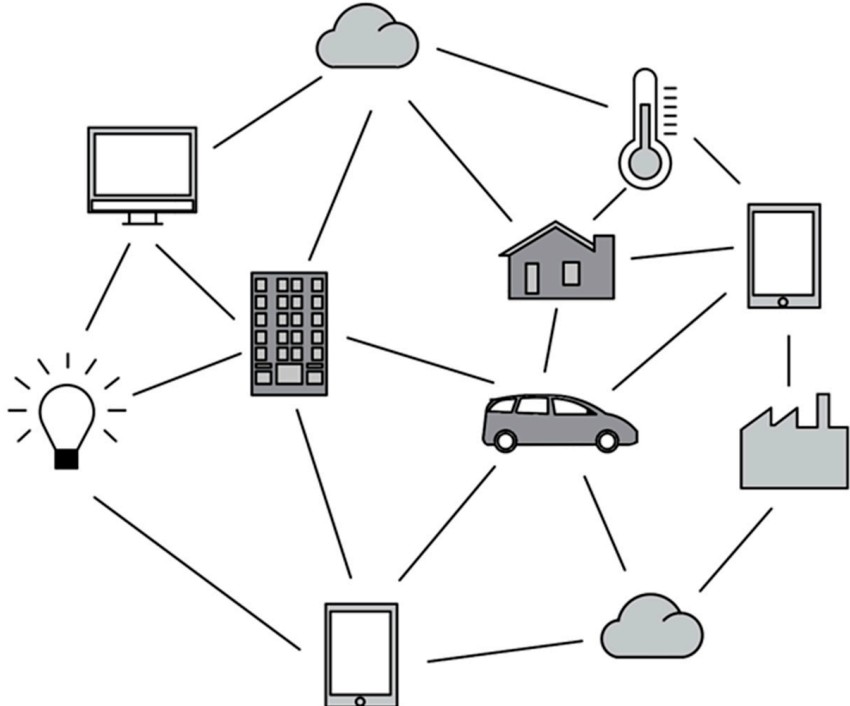

**Figure 1.** General IoT solution.

In addition to technical difficulties, the lack of a clear and widely accepted business model that can attract investments to support the deployment of these technologies is hindering the adoption of the IoT paradigm [5]. In this complex scenario, the application of the IoT paradigm in an urban context is particularly interesting because it responds to the strong push by many national governments to adopt ICT (Information and communication technologies) solutions in governance and, thus, implement the so-called Smart City concept [6]. Thus, IoT is often referred to as "a set of technologies enabling access to data deployed by various devices over wireless and wired Internet networks". Although there are differences in the definitions of the Internet of Things, there is consensus on the basic definition: It is the ability to deliver valuable and efficient information to different user devices via wireless and wired internet networks.

In line with the issues being addressed in this paper, a smart grid will be defined. It is a modern infrastructure that relies on ICT and digital networks to collect data about the power grid. For example, it collects data on electricity generation and consumption, which it then uses to build real consumption models to optimize the grid state in real-time.

Since the planning horizons for energy production and consumption can be short—for example, one hour ahead—advanced smart grid automation will generate a considerable amount of operational data in a fast decision-making environment. New algorithms will help make them adaptive and capable of reasoned prediction. On the other hand, new rules for managing, operating, and marketing the grids will be needed.

We encounter IoT when addressing the energy performance of buildings in the context of energy sustainability, reducing energy consumption and $CO_2$ emissions, which can now be considered as one of the fundamental society-wide challenges on the part of energy sustainability and the sustainability of the socioeconomic climate. Hence, the main focus of our research work is to create the conditions and prerequisites for the application of the Internet of Energy (IoE), or rather IoT, as IoE has a significant impact on the energy sector of smart cities [7]. IoE is the implementation of Internet of Things (IoT) technology into distributed energy systems and aims to achieve energy efficiency, prevent energy waste, and improve environmental conditions. At this point, the basic goals of our work can be set:

1. Perform analysis of IoT applied protocols and transmission systems.
2. Define and design the basic energy model of the building(s) with the IoT integration as an indispensable feature of this platform.
3. Design the smart building (smart homes, smart offices, etc.) template.
4. Experiment. Methodological procedure for solving the energy performance of buildings (reduction of energy consumption and $CO_2$).

Thus, in our case, IoT allows objects under distributed energy systems, such as a RES microgrid, to be detected or operated directly through existing grid technology, enabling a tighter connection between the real world and the computer network, increasing energy efficiency through system optimization [8,9]. Today, IoT is present on such a platform of applications that it is applicable in a vast field of science and technology [10]. It is mainly used in the fields of computing and informatics and big data processing, thus fulfilling the process conditions of a function of smart cities, namely mobile devices, charging stations for electromobility, smart monitoring, smart energy supply systems, process water supply, environmental safety, smart retail, smart supply chains, and online shopping [11,12]. Over the last ten years or so, IoT platforms and services have changed with the development and availability of open-source hardware and the emergence of new technologies. Users should further educate themselves on IoT's new principles and applications and familiarize themselves with the technologies available in the market, which they could then apply in their activities and, thus, freely implement new technologies [13].

If IoE and IoT include, among others, the use of smart sensors and the integration of renewable energy, they become a tool of legal science to serve the purpose of a smart city. In this study, the European Union's decisions to draw up the Rules for facilitating the transformation of existing cities into smart buildings, starting with existing buildings, can be relied on. Therefore, a smart building template is designed to control the performance of all technical systems through IoT technology in order to achieve energy efficiency. In addition, an automatic remote control method supported by a cloud-based interface is proposed to improve the certification of existing buildings in terms of energy performance. This method minimizes time-consuming procedures, especially big data storage. It reduces the energy consumption of each building in terms of the basic essence of a smart building, i.e., reducing energy consumption and $CO_2$ production.

This approach allows for long-term economic gains, reduced fuel import/supply costs, more efficient energy generation, and reduced emissions from renewable energy sources. When it comes to effective real-time monitoring and management of power chain data, this plays an important role in the process of increasing energy demand. Optimizing energy storage involves three primary parts:

(1) Energy sources including upstream refining processes;
(2) Energy transfer processes comprising energy storage and distribution systems;
(3) Energy demand in the building and transport sectors. Energy chain of suppliers and their distribution [14,15].

IoT uses sensors and networked systems to detect and transmit information in real-time for rapid assessment and effective decision-making. It will also enable the energy sector's transformation into a decentralized system, namely a smart and optimized elec-

tricity grid. The automation, integration, and monitoring of production or non-production processes through the Internet of Things-based systems as well as communication technologies and installed sensors represents the backbone of the whole issue of energy efficiency [16]. When focusing on a comprehensive analysis of building energy performance data, preferably through Building Information Modelling (BIM) applications and sophisticated algorithms, energy consumption trends on the consumer and defined application side can be tracked and, thus, the energy consumption over different time periods can be managed better [17–19].

The European Union considers energy efficiency an important way to reduce its dependence on fossil fuels, protect the environment more effectively, and save on energy costs. The Energy Efficiency Directive became one of the most discussed topics in Europe in the last year; however, the draft Roadmap to a Low-Carbon Economy also takes savings into account.

*Developments Up to Now and Expected Steps in the Field of Building Energy Performance*

In the EU, approximately 40% of energy is consumed in buildings. At the same time, the buildings are the source of 36% of $CO_2$ emissions. Currently, approximately 35% of buildings in the EU are over 50 years old, and more than 75% of buildings are unnecessarily energy-intensive, with only approximately 0.4 to 1.2% of them being renovated each year (depending on the country). Therefore, refurbishment of existing buildings can bring significant energy savings. It is also true for the Czech Republic, where the average age of houses with multiple flats is 52 years, and that of family houses is 49 years. Reducing the building energy performance can also bring other economic, social, and environmental benefits. It has an indisputable impact on the availability of heating and cooling–approximately 16% of the population live in energy poverty just in the Czech Republic. Investing in reducing the building energy performance also stimulates the economy, especially the construction sector, which accounts for around 9% of European GDP and directly employs 18 million people.

The term "Energy Performance of Buildings" (EPB) of 2018/844/EU is defined as follows: The energy performance of a building shall be determined on the basis of calculated or actual energy use and shall reflect typical energy use for space heating, space cooling, domestic hot water, ventilation, built-in lighting, and other technical building systems.

The energy performance of a building shall be expressed by a numeric indicator of primary energy use in $kWh/(m^2.y)$ for the purpose of both energy performance certification and compliance with minimum energy performance requirements. The methodology applied to determine the energy performance of a building shall be transparent and open to innovation.

The amendments to Directive 201/31/EU clarify and add some terms in the definitions and interpretation of terms section. The term "technical building system" includes the technical equipment for space heating, space cooling, ventilation, domestic hot water, built-in lighting, building automation and control, and on-site electricity generation, or a combination thereof, including those systems using energy from renewable sources, of a building or building unit. Thus, the definition of the technical system is extended to include on-site electricity generation and automation and control systems, which are further defined as systems consisting of all products, software, and engineering services to provide automatic control or facilitate manual control.

In the next decade, it will be important to make efforts regarding the energy sustainability and energy performance of buildings, which are as follows:

- Further reduce greenhouse gas emissions–40% fewer greenhouse gases should be produced in 2030 compared to 1990, further promoting low greenhouse gas technologies.
- Create a sustainable, competitive, secure, and decarbonized energy system in Europe by 2050 (based on the claim that buildings are responsible for 36% of all emissions in the Union).

- Significantly increase energy efficiency efforts (and achieve a 36–37% reduction in energy consumption and a final reduction in primary energy consumption by 39–41%) by 2030 compared to the current main goal of at least 32.5%.

In terms of a long-term building renovation strategy, the goal is to achieve an 80–95% reduction in the Union's greenhouse gas emissions by 2050. For major renovations of existing buildings, it is recommended to focus the solution on the usage of technically and economically feasible highly efficient technical systems and the issues of a healthy indoor environment, fire safety, and risks associated with intense seismic activity.

Partial and preliminary data for 2020 show that the impact of the COVID-19 crisis on energy consumption is significant, and, as a result, the energy efficiency goals for that year can be easily met. However, these reductions are not due to structural changes. Moreover, before the crisis, it was clear that Member States' energy efficiency efforts alone would not be sufficient to achieve the set goals. The subsequent recovery from the COVID-19 crisis is expected to lead to a return of energy consumption to similar levels as before the crisis.

The new emphasis is on building automation from several perspectives. The installation of self-regulating devices for individual temperature control in each room is recommended, and all buildings except residential buildings should be equipped with an automation system since 2025. Other measures include the promotion of smart-ready systems that will enable the use of smart grids, more accurate information on actual savings achieved and more accurate data on consumption patterns.

The level of smart equipment and readiness of a building will now be expressed by an optional "smart readiness indicator" (SRI). The precise definition of the SRI and the method for its determination will be based on the ability of a building and the technical systems thereof to maintain the energy performance level and energy-efficient operation of the building by adapting energy consumption, for example, by using renewable energy; adapting its operating mode in response to the needs of users with due consideration of user-friendliness; maintaining a healthy indoor environment and the ability to report on energy use and the flexibility of the building in terms of overall electricity needs and the ability to shift loads over time.

The development and promotion of electromobility technologies have translated into requirements to create charging infrastructure respecting local technical and economic conditions. For new and renovated residential buildings with more than 10 parking places, wiring shall be provided for each of them. The Directive addresses in detail the conditions and exceptions for applying these rules. The timing of checks for fulfilling these requirements suggests that this is a long-term process.

The new European Directive on the energy performance of buildings is a further step on the path towards reducing energy consumption in buildings. It is important to note that this Directive does not have a direct effect but must be translated into national laws and regulations. At the same time, it should be noted that this is not a legally binding interpretation claiming to be complete but a basic building block to be relied upon when addressing this issue on a research and development platform. Therefore, the above claims, conditions, and recommendations are reflected in the design of our research on the platform of the experiment being presented in this paper.

Last but not least, it is important to recall that the requirements of Directive 2010/31/EU require, among other things, that the design documentation for all new buildings has to meet the requirements for a so-called Nearly Zero Energy Building ("NZEB") since 1 January 2020. A nearly zero energy building is a very low-energy building, where the energy consumption is covered largely by renewable sources (Act No. 406/2000 Coll., on Energy Management, as amended).

As mentioned earlier, our research aims to design a smart building template that controls the performance of all technical systems via IoT technology with regard to achieving energy efficiency and sustainable energy within a smart district, or more precisely, a smart city system.

Smart building (home) equipment can be divided into the following areas according to the benefits they bring to home users [20–22]:

- Comfort
- Assistive technologies
- Security
- Energy saving

New technologies allow the creation of buildings that are more efficient in terms of energy and water consumption, or generally use more efficient and sustainable resources. Buildings with high-performance appliances that significantly reduce resource usage are designed. Features that allow the building to generate more energy than it consumes are included, smart meters are installed, the Internet of Things is implemented, and large amounts of data on the efficiency of the building's operations are analyzed. Today, so-called smart buildings are created. The building is a part of a larger city system. The city-systems are very complex systems with the following constantly interacting system components: the man-made environment (buildings, infrastructure, urban planning, etc.), its social environment based on paradigms, values, policies and available technologies, and the natural environment the system needs to exist. These components of the city system are constantly interacting with each other, affecting the city's functioning, surroundings thereof, the operation of buildings in the city, and the people who live, work, and spend their leisure time there [23].

A smart city is not just a collection of smart buildings connected to other smart buildings and decentralized microsystems. It is a complex system managing a constant supply and demand, a continuous relationship between giving and receiving, and continuous exchange among its built environment, social environment, and natural environment. A smart city can only work if all the system components thereof take advantage of this constant flow of exchange, providing information and feedback loops on the necessary inputs and resulting outputs. A smart building is not only an energy consumer but also a producer. A smart building is a power plant within a larger energy-intensive and manufacturing subjects system. The conventional idea of delivering public utilities and requiring them by the building is obsolete in the era of smart buildings. In the era of smart buildings, a new grid model is needed. Grids are no longer the main energy producer. Their main task is to manage multiple sources of energy production and distribute them correctly. Everyone in the system becomes a micro-power plant and the network administrator is responsible for its management. We can show this in Figure 2, although this situation is well known today.

Smart cities are a system of multiple sources of energy generated by the users themselves, as well as grids, the main task of which is management and distribution [24]. In order to manage a smart grid, it must be divided into several decentralized microgrids.

The decentralized microgrids will also improve transmission and distribution losses. About a quarter of all energy is lost on the way from the power plant to the consumer. The smaller the system, the shorter the distance between supply and demand and the fewer losses may occur during distribution. The grid itself will become a means of energy storage for anyone having the energy to feed into it. In addition, the decentralized systems are much more reliable and resilient. In case of outages, the system may use neighboring systems to serve as a backup. Large, centralized systems are dependent on one main system and must install several additional backup systems operated redundantly.

Our goal globally is to reduce energy consumption in a way that significantly reduces the negative impact of our consumption on the environment and, thus, on nature (including ourselves). In order to achieve this fundamental goal, engineering and automation of integrated building systems have to be focused on. Logically, integrating and operating these buildings to reduce energy consumption is a top priority. It is possible to achieve this only by integrating the system and using the powerful technologies available today to ensure the facility's efficient operation. For this reason, these technologies can be used to create integrated systems. Combined with new technologies such as Cloud and IoT, more

efficient buildings that use current technologies to reduce consumption can be created. Building automation is evolving to meet the need to reduce energy consumption.

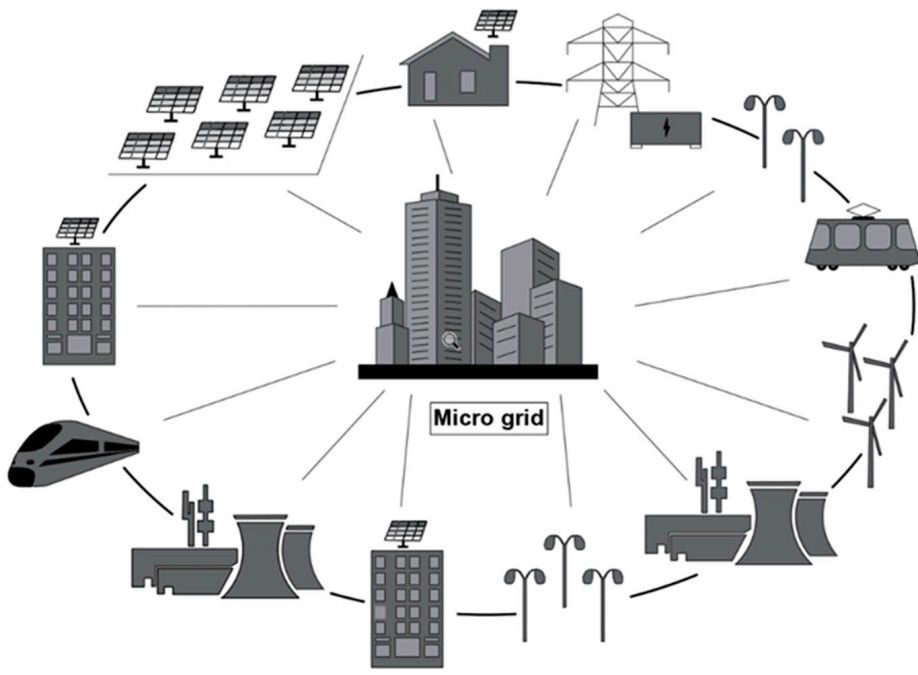

**Figure 2.** A smart microgrid as part of a cluster of buildings.

However, the building automation industry has continued to work hard to develop solutions addressing the construction market and reduction in the energy consumption of buildings. The end-user has started to ask about the interconnection of various systems to allow for a competitive choice of products and service providers. Open protocols have begun to become a common topic of discussion as users realize the power of integration. During this time, our world begins to see the impact of our energy consumption on the planet. However, we have continued to develop solutions that dramatically reduce our impact: open protocols for fully integrated systems. This was the beginning of the BACnet® Project Standards Committee in 1987 (introduced as an ANS/ASHRAE standard in 1995) and the introduction of LonWorks® in 1990 by Echelon Corporation (submission and adoption of LonTalk® as an ANSI standard in 1999). The Internet of Things has opened up a whole new world of interconnectivity, enabling the use of smart mobile devices and apps for these complex systems. Web browsers, SaaS (Software as a Service), and mobile applications can be used to create a dynamic system where data storage capacities are virtually unlimited, and access and control are within reach. With all these options, the efficiency is greater than ever. However, success can only be achieved through successful integration. This led us to set another goal for our research: to analyze issues related to integration and protocols. In this case, it was confirmed that it was important to evaluate which devices could be integrated and which could not. The good news in these situations is that many manufacturers are now starting to create native or input devices allowing for various systems to be connected using open protocols. Some practical examples of the application of Internet communication to facilitate the process of technical equipment automation in buildings and their coordination within smart cities, where the main purpose is sustainable energy in the system of the energy performance of a building, will be shown.

## 2. Materials and Methods Background

As mentioned in the introduction, we are concerned with defining and designing a building energy model (BEM) with IoT integration. In this case, the defined assessed locality of the Prague 6–Dejvice district (enclosing the area of Vítězné náměstí), as shown in

Figure 3, will be presented. An experiment will be carried out on this urban area to address the smart, sustainable energy industry aiming to reduce energy consumption and $CO_2$ emissions when applying EMB, IoT, and building control systems. The buildings in the said urban area will have the character of NZEB after reconstruction.

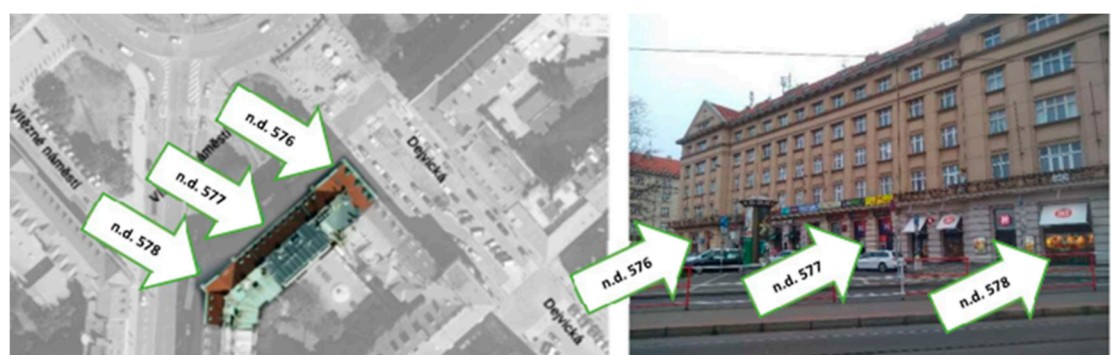

**Figure 3.** Designation of buildings (their descriptive numbers), where the energy performance of buildings was solved in the experiment, Prague 6-Bubeneč.

In this case, the so-called "Smart Energy System" (SES) can be talked about. The Smart Energy System (SES) consists of:

(a)　Smart integration of decentralized sources of sustainable energy (RES microgrid, smart grid), see Figure 4 (highlighted in green);

(b)　Efficient distribution (Cloud computing), see Figure 5 (represented by the "CLOUD");

(c)　Optimized energy consumption and $CO_2$ emission reduction (building control system applications, EXAMPLE: KNF/FOXTROT etc.).

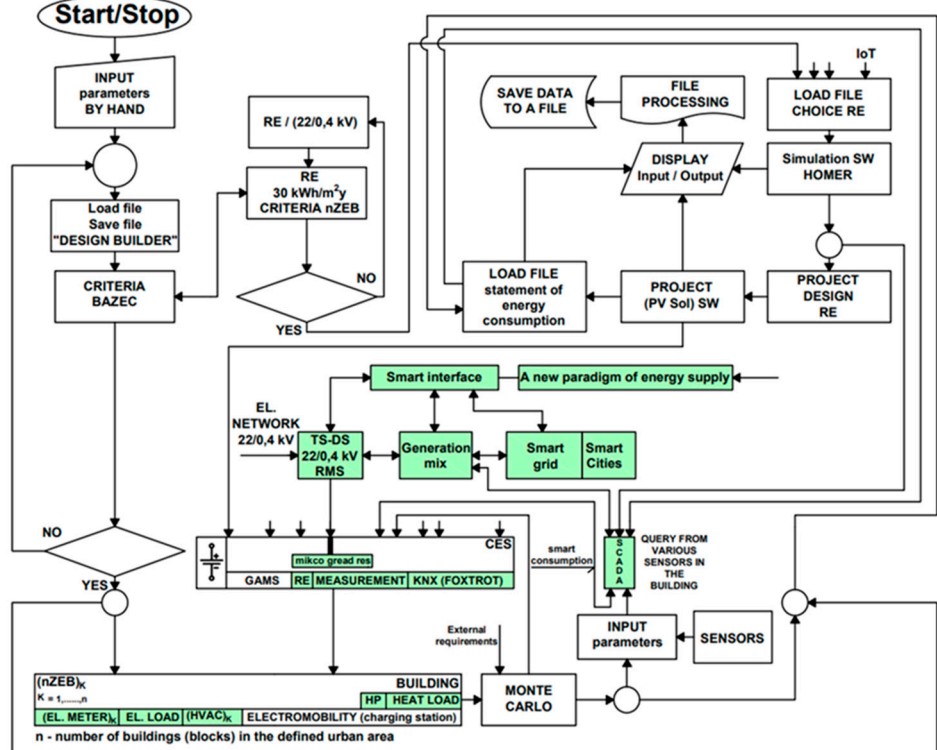

**Figure 4.** NZEB energy model with the application of automated systems of technical equipment of buildings.

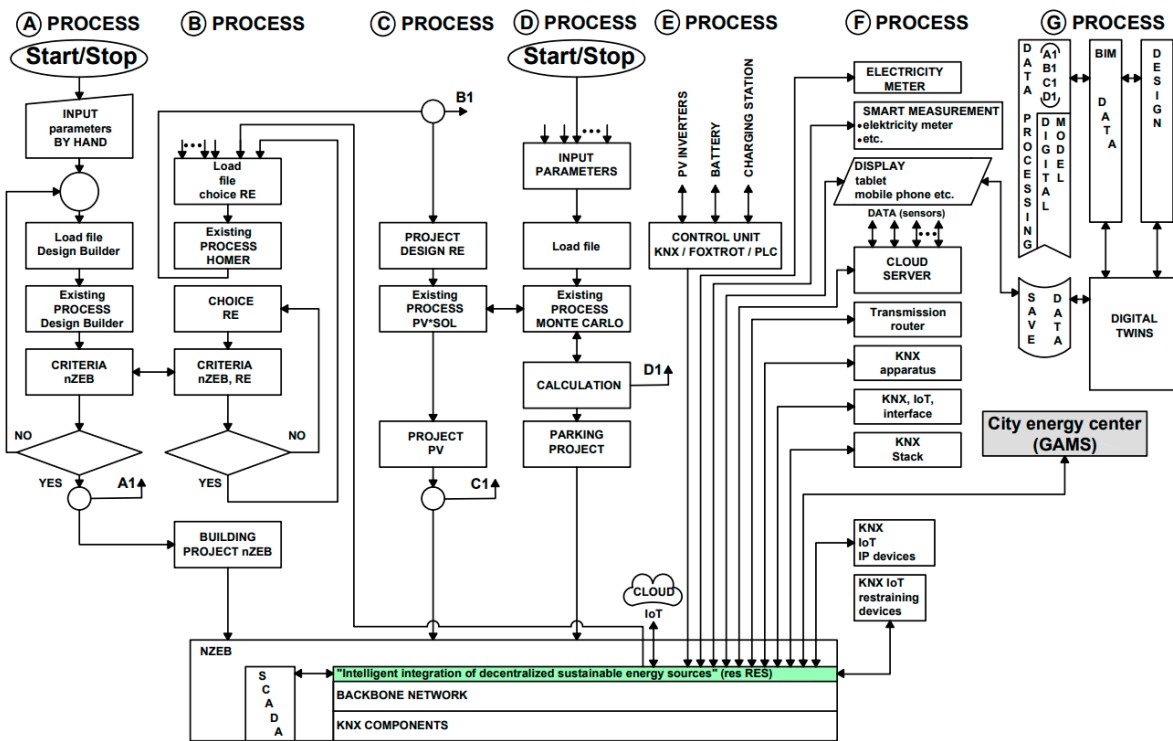

**Figure 5.** IoT with integration of the NZEB control system in the BEM system.

## 2.1. SCADA Service, Energy Consumption in Existing Buildings

This is a kind of dispatching system (unique interface) that allows you to keep informed about your projects from anywhere in the world, with a free service to monitor and manage the operation of all technologies that are connected to SCADA. The dispatch interface can then be accessed remotely from multiple devices. SCADA will allow you to maintain an overall view of the system status, respond to timely crisis situations, optimize operations, facilitate problem solving by analyzing historical data, and reduce operating costs due to fewer demands on infrastructure and staff.

Furthermore, visualization is known in the building industry as SCADA (Supervisory Control And Data Acquisition-systems for industrial control and data collection). This is software with which it is possible to concentrate all technologies, not only KNX/EIB, into one unit and thus have control over everything that happens or happens in the building. KNX/EIB means an internationally standardized system of programmable electrical installations. It is an open system; any other stand-alone system can be integrated into it. The set of standards dealing with intelligent electrical installation is ČSN EN 5009. Therefore, thanks to these and other management skills, we have included it in Figure 4, where it performs functions that support the importance of NZEB.

Therefore, smart energy consists of three independent building blocks that must be interconnected and communicate with each other effectively to form a unified "smart energy system". These building blocks are made up of (see Figure 5):

- Block 1, comprising process A, B (DesignBuilder, HOMER);
- Block 2, comprising process C, D (application of PV*SOL and Monte Carlo simulation programs);
- Block 3, comprising process E, F (application of IoT and KNX, PLC Tecomat Foxtrot, ABB-free@home® control system).

Note that: PLC Tecomat Foxtrot is actually a compact modular control and regulation system for small and medium-sized applications of technical equipment in buildings and ABB-free @ home® is an intelligent wiring system for modern living.

Figure 6 shows the procedure of the EPB solution according to the EMB in steps from the input parameters (data) (indicated as 1 in Figure 6) to the simulation software DesignBuilder (DB). DesignBuilder (indicated as "A" in Figure 6) is software for complex dynamic building modelling, analysis, and environmental assessment. The main purpose of this software is to calculate the energy consumption of a building. The building model can be imported from other BIM programs. To perform the simulation, two main components of the energy model of buildings need to be created/modified:

- Building materials and components (walls, floors, ceilings, occupants, and equipment);
- Equipment components (HVAC equipment and other environment-control systems).

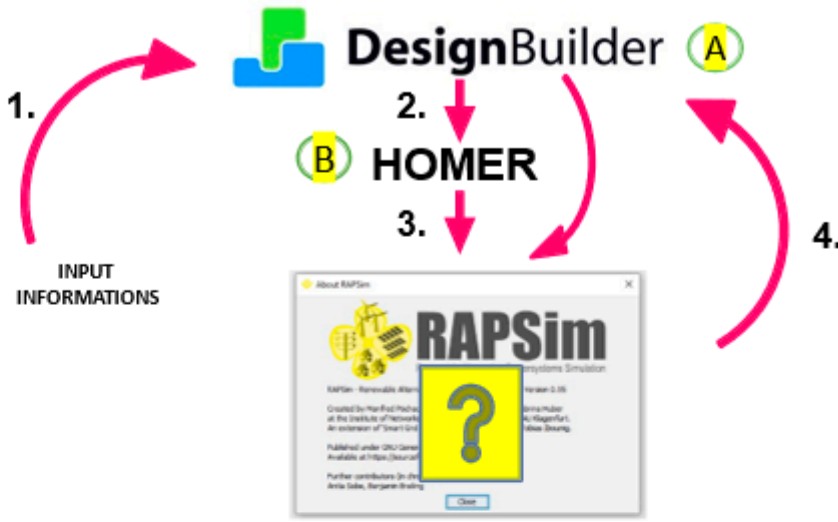

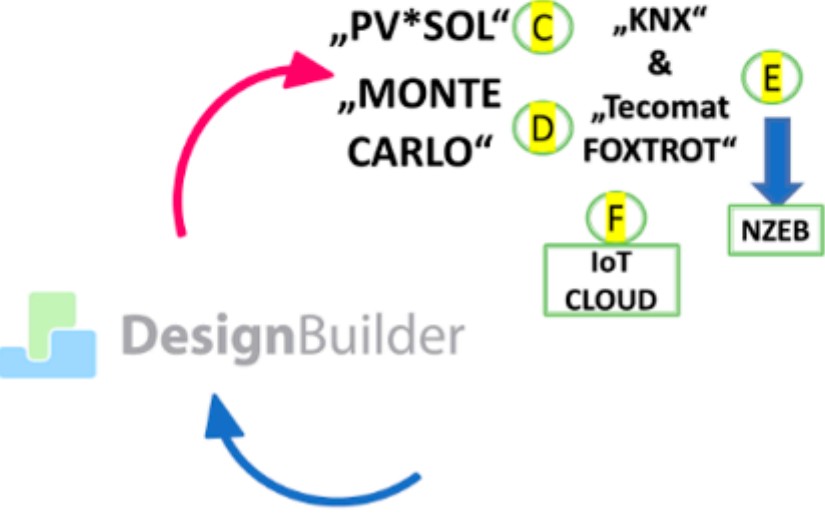

**Figure 6.** EPB Solution Principle as a Unified SES.

The input data (indicated as 2 in Figure 6) include: The value of the heat transfer coefficient of the structure and openings, the conductivity ratio of the solar device, and the building occupancy; after simulation, it is possible to obtain building and block output of zonal data (average temperatures, comfort conditions), internal gains, and latent loads.

After modelling and calculations, the output data will be obtained as heat loss, solar energy, room temperature, amount of energy needed for heating, ventilation, and air conditioning.

After adjusting the display of the results, a comprehensive analysis and comparison of the energy characteristics of the building can be performed (see Tables 1 and 2). In the settings, the user can choose which variables are included in the display of the internal boost, fuel degradation, and comfort or fuel graphs, including $CO_2$ values. All this information can be displayed as a graph, table, or summary.

**Table 1.** Reference values for EPB and NZEB for countries in different EU climate zones EU (Commission's Recommendation (EU) 2016/1318 of 29 July 2016 on guidelines to promote nearly zero-energy buildings and best practices to ensure that all new buildings are nearly zero energy by 2020).

| Climate Zone (From the Studies "Towards Nearly Zero-Energy Buildings–Definition on Common Principles under the EPBD" by Ecofys: | Administrative Buildings | | | New Family Houses | | |
|---|---|---|---|---|---|---|
| | Net Primary Energy per Year [kWh/m$^2$] | Primary Energy Consumption per Year [kWh/m$^2$] | Coverage by RES per Year [kWh/m$^2$] | Net Primary Energy per Year [kWh/m$^2$] | Primary Energy Consumption per Year [kWh/m$^2$] | Coverage by RES per Year [kWh/m$^2$] |
| Mediterranean | 20–30 | 80–90 | 60 | 0–15 | 50–65 | 50 |
| Oceanic | 40–50 | 85–100 | 45 | 15–30 | 50–65 | 35 |
| Continental | 40–55 | 85–100 | 45 | 20–40 | 50–70 | 30 |
| Nordic | 55–70 | 85–100 | 30 | 40–65 | 65–90 | 25 |

Climate Zone descriptions:
- The Mediterranean Is Specified as Zone 1: Catania (Other Cities: Athens, Larnaca, Luga, Seville, Palermo);
- Oceanic Is Specified as Zone 4: Paris (Other Cities: Amsterdam, Berlin, Brussels, Copenhagen, Dublin, London, Macoun, Nancy, Prague, Warsaw);
- Continental Is Specified as Zone 3: Budapest (Other Cities: Bratislava, Ljubljana, Milan, Vienna, Prague);
- Nordic Is Specified as Zone 5: Stockholm (Helsinki, Riga, Stockholm, Gdansk, Tovaren).)

**Table 2.** Normative Values of the Heat Transfer Coefficient (CSN 73 0540-2: 2011 Thermal protection of buildings–Part 2: Requirements).

| Description of Structure | Heat Transfer Coefficient [W/m$^2$.K] | | |
|---|---|---|---|
| | Required Values UN,20 | Recommended Values Urec,20 | Recommended Values for Passive Buildings Upas,20 |
| Outside wall | 0.30 | Heavy: 0.25 Lightweight: 0.20 | 0.18 to 0.12 |
| Flat and pitched roof with a pitch up to 450 inclusive | 0.24 | 0.16 | 0.15 to 0.10 |
| The floor and wall of the heated space is adjacent to the soil | 0.45 | 0.30 | 0.22 to 0.15 |
| The opening panel in the outside wall and pitched roof, from the heated space to the outside environment | 1.5 | 1.2 | 0.8 to 0.6 |

Next, Figure 6 is followed to comply with the regulation according to Table 1 (RES coverage: 35 kWh/m$^2$.r) using the HOMER software application (Hybrid Optimization Model for Electrical Renewable), marked with the letter "B". This software application develops and evaluates options for autonomous and networked power systems technically and financially for remote, autonomous, and distributed applications. The software may cover solar photovoltaic systems, wind turbines, diesel generators, inverters, and batteries. It will allow a large number of technological options to be considered, taking into account

energy availability and other variables. The optimized results of the microgrid diagram (see Figure 7) are shown in the output (indicated as 3 in Figure 6). They are ranked by Cost of Energy from lowest to highest. This section is a result of the specification of a given microgrid design.

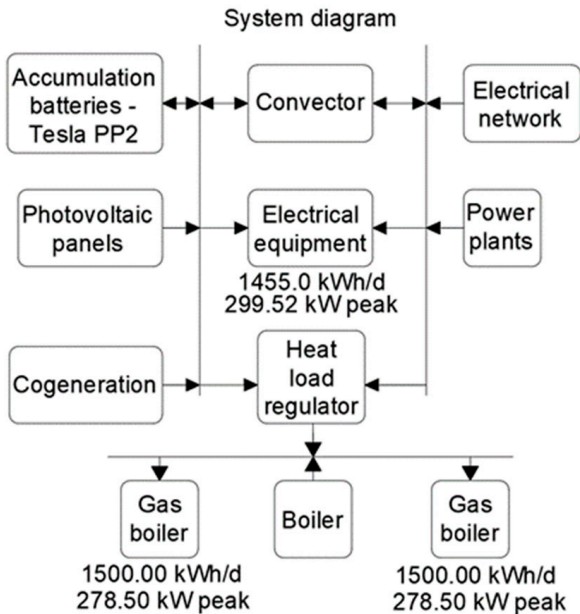

**Figure 7.** RES Microgrid Design.

Note: The simulation software RAPSIM is not considered.

Next, based on the result of the DB simulation output, the solution of the photovoltaic system (PVS), which is solved by the simulation of PV*SOL software (indicated as "C" in Figure 6), follows within the context of the HOMER output. The program is used to design photovoltaic panels and inverters, and to determine the distribution of annual electricity consumption of given objects and assess the need to install battery storage on site.

Since the area of the block of flats under consideration has parking places, the energy load of a given area must be addressed regarding installation position for charging stations in the building cluster under consideration (smart urban area). For this purpose, it is necessary to consider what the average electricity consumption is in the given area. According to Figure 6, this issue is solved by Monte Carlo software simulation (indicated as "D" in Figure 6). This is a stochastic heuristic class of algorithms for simulating systems using pseudorandom numbers. This method is based on the determination of the mean value of a quantity through a random process.

Finally, the EPB solution according to BEM is as follows: community energy is solved by IoT and IoE applications (indicated as "F" in Figure 6), while the energy systems of technical building equipment are controlled by KNX, FOXTROT or ABB-free@home® (indicated as "E" in Figure 6). The result of this complex process of EPB solutions according to EMB—see Figures 1 and 2—is a building (cluster of buildings) or a smart building (smart urban area), and thus, a basic element of Smart Cities.

In the described case of the solution or reconstruction of new or existing buildings, these buildings within the specified area of Prague 6 are solved on the NZEB platform. The emphasis is on sustainable energy, which has the greatest potential to reduce $CO_2$ emissions using the change of sources by which the urban area (city) under consideration de facto covers the electricity consumption thereof. Using newly built solar and other emission-free power generation plants, electricity supply can be secured without coal power plants by 2030. This will reduce carbon emissions by approximately 2.5 million tons (22.5%) in Prague (https://klima.praha.eu/cs/udrzitelna-energetika-a-budovy.html, accessed on 4 March 2022). This is despite the expected growth in electricity consumption

due to the development of electromobility and the increased electrification of the heating and cooling sector.

### 2.2. Distributed Systems and Cloud Computing

The emergence of distributed systems (DS) has brought new challenges in scheduling in computer systems, including clusters, grids, and, more recently, clouds. The distributed systems are all around us, and their importance and complexity are growing. The distributed system is a collection of independent, autonomous computing elements connected via a communication network. The computing elements communicate by sending messages to collaborate in some way. When applying IoT in DS, computers work concurrently, do not share a global clock, and fail independently. Dependencies among individual DS computers amplify the consequences of failure. For example, there are $n$ dependent processes, the probability of failure is $p \rightarrow (1-p)^n$. The question arises: Why are there DSs? Because the so-called inherent distribution is the crux of the matter. Applications requiring resource sharing or information dissemination among geographically or organizationally foreign entities represent "natural" distributed systems. The DS application aims to be able to solve more tasks or larger instances of tasks than is possible with a single computer, as in parallel computing. Another characteristic of DS is openness, security, etc., as well as the ability to ensure (almost) permanent availability of the required services. For parallel computing, this goal is not set typically.

The distributed system generally has the following basic characteristics:

(a)　Resources can be shared just like software from other systems connected to the network: in other words, the components in the system are synchronous [25].
(b)　A global clock is not necessary to have in DS.
(c)　In the distributed model, the error tolerance is much higher than in other network models; otherwise, the performance/cost ratio is quite interesting [26].

The basic objectives of the DS are:

(a)　Transparency;
(b)　Openness [27];
(c)　Flexibility;
(d)　Reliability [28];
(e)　Performance;
(f)　Scalability [29–31].

The distributed system may face virtually any problem, such as security; this creates room for additional solutions, especially when using a public network. Another distribution problem is fault-tolerance if it is built on an unreliable foundation [32]. Without proper protocols or principles, resource coordination and sharing represent a big challenge in a distributed environment [33].

Providing computing performance through the network (Internet) is advantageous for cloud computing. Clients do not need the services of massive, complex computers to trade, track, and manage assets and devices; rather, they can access cloud storage services depending on their needs. Simply put, cloud computing is the delivery of computing services, including servers, storage, databases, networks, software, analytics tools, and intelligent functions, through the Internet (the "cloud"), offering faster innovation, resource flexibility, and cost advantages.

According to the scope of services provided within cloud computing, it can be divided into several groups [34]:

- Infrastructure as a Service (IaaS)–The cloud provides a complete infrastructure (most often virtualization). A provider is fully responsible for the service provision and management.
- Platform as a Service (PaaS)–The cloud provides resources for the entire lifecycle of a web application's creation and delivery.
- Software as a Service (SaaS)–A user buys or rents access to the application, not the application itself.

- IBM further divides cloud computing according to how the cloud itself is provided [29].
- Public cloud—This is provided by companies that offer fast access to computing resources through a public network. The user does not have to purchase their own hardware or software.
- Private cloud—The infrastructure is operated exclusively for a single organisation.
- Hybrid cloud—This is a combination of public and private clouds.
- The most well-known cloud computing platforms include:
- Google Apps (SaaS), which covers, e.g., email, docs, Gtalk, calendar, etc.
- Microsoft 365 (SaaS) covering documents, websites, and video conferencing.
- Amazon EC2 (PaaS) is part of Amazon Web Services (AWS, offering virtual machine rental).
- IBM Cloud (SaaS, PaaS, IaaS), which mainly covers commercial services.

Analysis of the Applied Protocols and Transmission Systems within IoT

The Internet of Things uses a wide range of possible protocols to transfer data. These protocols have been derived from telecommunication protocols. The main feature of these protocols consists in the transmission of small data volume (in the order of tens to hundreds of bits), so they do not have large energy and memory requirements. These protocols belong to the TCP/IP protocol family. It contains four basic layers: the application layer, transport layer, network layer, and network-interface layer [35].

Figure 8 shows the protocols used in IoT. Let's take a closer look at four protocols belonging to the application layer and one that belongs to the datalink layer. The most important protocols that are most used in IoT will be presented in the following section. These selected protocols try to work with the data in various ways, but standardizing allows for a "thing" to recognize how the data should be further used or displayed to the user. Therefore, these protocols allow for "things" to communicate seamlessly among various kinds of devices. Below, the basic protocols and transmission systems are listed:

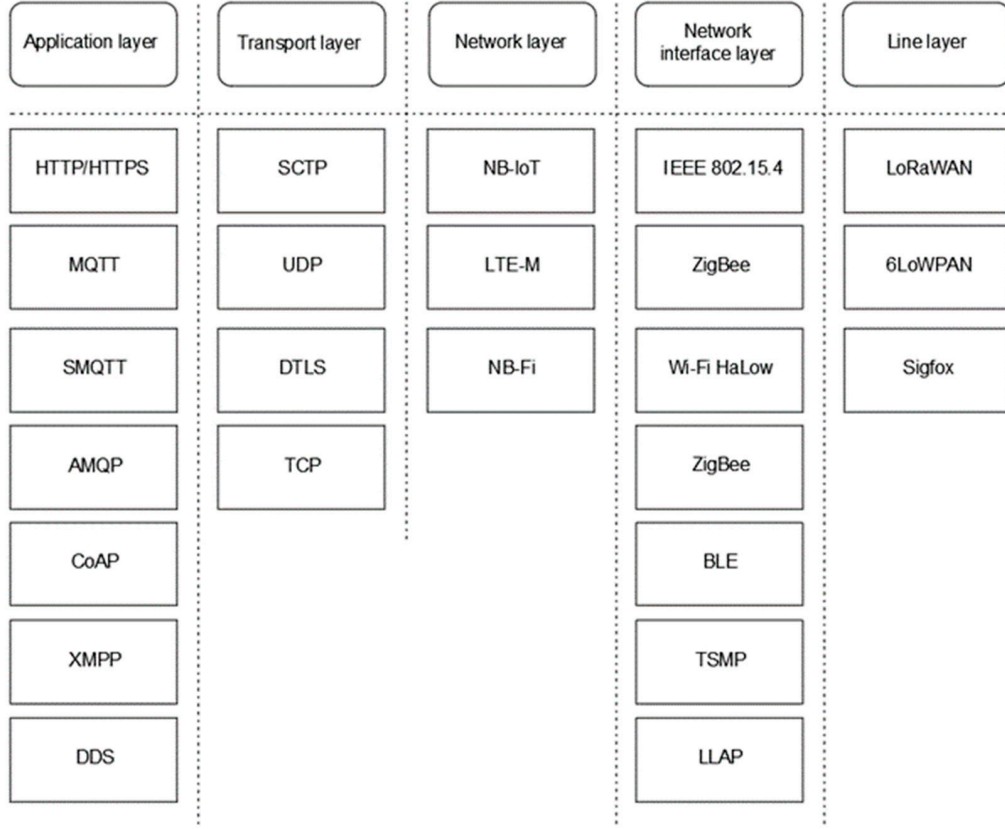

**Figure 8.** IoT protocols and standards.

1.   HTTP

HTTP (Hypertext Transfer Protocol) is the most well-known protocol today. It allows communication between client and server via TCP (Transmission Control Protocol). It is the most widely used transport layer protocol in the TCP/IP suite of protocols used on the Internet. It is used to transfer hypertext documents in HTML, XML, and other formats. This protocol provides the transfer of information between endpoints. In 2015, a new version called HTTP/2 was released, featuring better compression and data transfer. This makes it much faster than the older version [35,36].

2.   MQTT

MQTT (Message Queuing Telemetry Transport) represents a standard protocol in IoT as well. It forwards messages between devices using a central point, called a broker. The message content can be arbitrary; however, it is limited by the size defined by the protocol version. The transfer in the protocol is carried out using TCP/IP and works with the publisher and subscriber models. The messages are sorted into topics and published on the given topic. Based on this, the device sends the message to the central point, which stores it and makes it available to other devices. Then, other devices subscribe to these messages. The protocol was created by IBM in 1999. In 2013, it was standardized by OASIS (Organization for the Advancement of Structured Information Standards) [35] and MQTT (undated).

3.   AMQP

AMQP (Advanced Message Queue Protocol) is a protocol for passing business messages between applications or organizations. It is considered an asynchronous addition to the HTTP protocol. It is not a normal publish/subscribe protocol; it rather works with MOM (Message-oriented middleware). Message oriented middleware (MOM) is a type of software product that enables message distribution over complex IT systems [37]. The protocol defines a set of rules that must be followed in order to transfer a certain amount of bytes. AMQP was founded in 2003 when the first version of the protocol was released. AMQP has been a member of OASIS since 2011.

4.   CoAP

CoAP (Constrained Application Protocol) is a protocol for constrained devices and constrained networks in IoT. Constrained devices are called nodes. It is based on the HTTP protocol but differs from it in many ways. It uses the UDP (User Datagram Protocol), which is a transfer protocol, unlike HTTP, which uses TCP. By this, the burden of maintaining a connection is removed; it is advantageous for smart devices having limited capabilities. To reduce overhead, CoAP uses a binary format instead of the text format used by HTTP. Messages are exchanged between UDP endpoints in CoAP. CoAP was created by the IETF (Internet Engineering Task Force) CoRE (Constrained RESTful Environments) and became an Internet standard in 2014 [35].

5.   6LoWPAN

6LoWPAN (Low-power Wireless Personal Area Networks) is a protocol that falls into the datalink layer of IoT. Within this network, communication takes place via the IPv6 network protocol. The protocol has a lower baud rate as well as lower operation cost [35]. The fourth version of the Internet Protocol, IPv4, will not cover the range of public addresses assigned to objects in the future. Therefore, IPv6 has been created to provide more address space. Compared to IPv4, the address is 128-bits long and, thus, approximately $3.4 \times 1038$ unique IP addresses can be assigned.

6.   Transmission Systems

All data measured, read, or entered by a user must be transmitted to a central collection system for further processing. A suitable communication system is required for this transmission, most often wireless technology. There are many communication technologies

suitable for IoT. These are mainly wireless technologies that allow connections from a few centimeters (RFID) to hundreds of kilometers (Sigfox, LoRa). Important parameters for choosing the right technology include range, data transmission rate, energy consumption, and security.

Within Smart City, LPWANs (Low-PowerWide-Area Networks) are predominantly used, including Sigfox, LoRa, and NB-It. These networks are energy-efficient and have a range of tens of kilometers. Short-range networks include RFID, Bluetooth, ZigBee, and Z-Wave. Medium-range networks include IQRF and WiFi. Long-range networks include Sigfox, LoRa, NarrowBandlo T, GPRS, HSDPA (3G), LTE (4G), and 5G.

### 2.3. Basic BEM Design with IoT Integration

The Internet of Things (IoT) is at the heart of smart urban area (city) implementation. In other words, IoT represents the intelligent technical backbone of smart urban areas (cities), as shown in Figure 9. Smart "urban areas" (cities) must have three key characteristics:

- Intelligence,
- Network connection,
- Instrumentation, that IoT can provide. It can be said that by using IoT, smart areas or cities will become feasible. The use of smartphones, smart meters, smart sensors, and radio frequency identification (RFID) essentially forms the framework of IoT in smart cities (see Figure 10) [38].

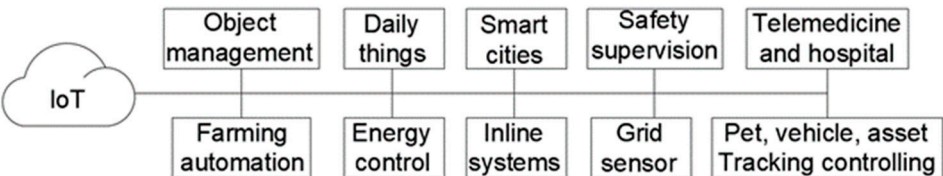

**Figure 9.** Smart Cities and IoT.

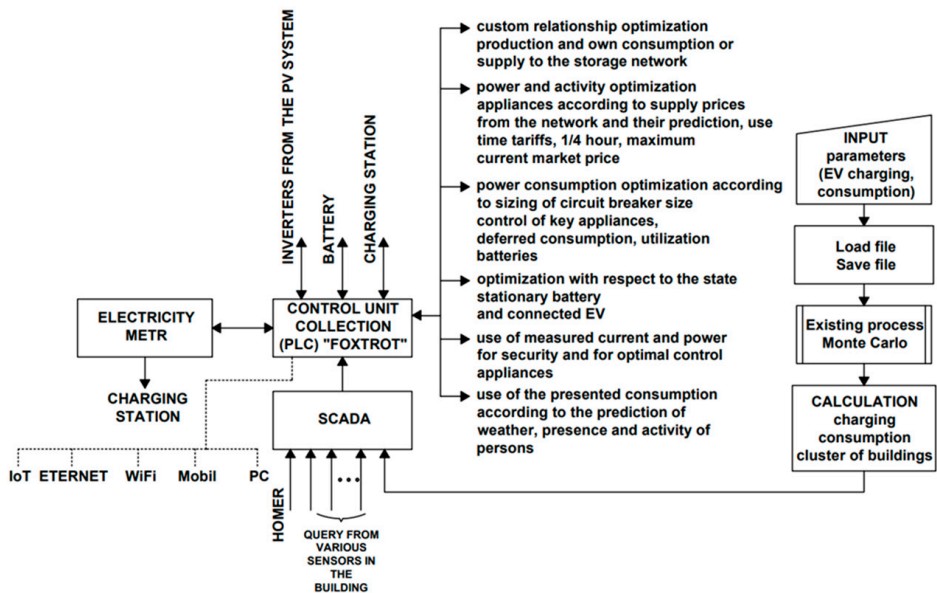

**Figure 10.** Optimization of energy consumption, IoT and building management system.

In addition to the key characteristics of a smart urban area, or more precisely a smart city, the smart energy system (SES) is a compatible and integrating element. "A smart energy system consists of the smart integration of decentralized sources of sustainable energy, efficient distribution, and optimized energy consumption. Therefore, smart energy consists of three independent building blocks that must be interconnected and communicate with each other effectively to form a unified smart energy system" [39]. The fulfilment of

the SES is achieved by proposing a new approach to the solution thereof, which is shown structurally (model) in Figures 4 and 5. The solution for energy consumption optimization is documented in Figure 10. Subsequently, the SES is verified in our experiment, which will be described in the following section.

The sensor (see Figure 10) can be considered the lowest layer of the IoT architecture that allows collecting data from the environment and sending it to SCADA for further processing. The sensors can be integrated into various electrical appliances to create an IoT-integrated device. These "smart" appliances can be remotely controlled and managed by a PLC (Programmable Logic Controller) or FOXTROT/KNX/ABB-free@home®, and they can read information. They can be connected via structured wiring or wireless communication via a cloud. Communication can be carried out directly from the device to the cloud (device-to-cloud), or the device communicates with a gateway connected to the Internet (device-to-gateway). Typically, the gateway includes software facilitating the data transfer to the cloud application.

### 2.3.1. The KNX IoT Specification Adds the Following Additional Options

Because of the KNX IoT based project, more KNX application options have been added. See Figure 11 for a brief summary of the system's functions. If it is an application of KNX IoT, we will show the specification of KNX IoT as another possibility of application in the framework of energy sustainability.

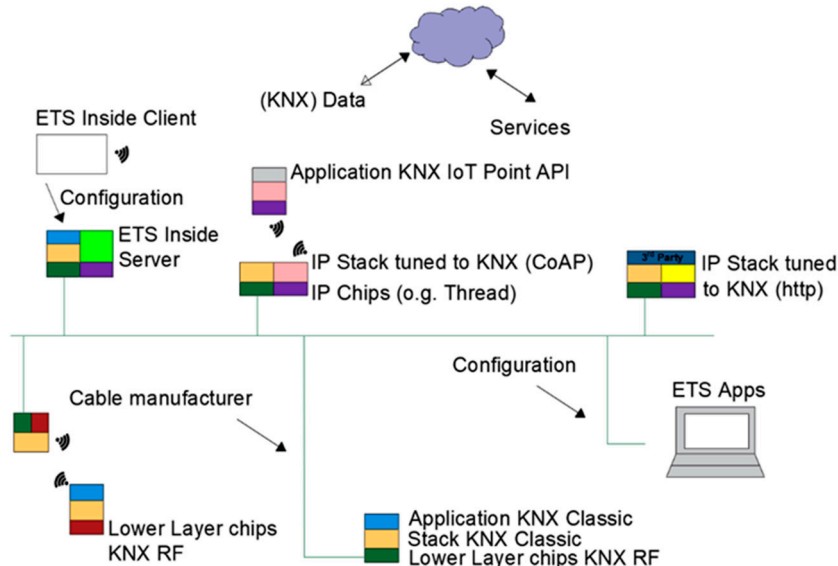

**Figure 11.** KNX-IoT system.

### 2.3.2. Third Party's KNX IoT API

The third party's API (Application Programming Interface) has been designed to support WebSocket interfaces, allowing, for example, direct cloud integration with two-way communication. WebSocket is a computer communication protocol providing a full-duplex (two-way) communication channel through a single TCP connection. The WebSocket protocol was standardized by the IETF as RFC 6455 in 2011, and the WebSocket API in Web IDL was standardized by the W3C consortium (https://cs.wikipedia.org/wiki/WebSocket, accessed on 4 March 2022).

Thus, the cloud applications can subscribe to on-demand information, minimizing data transfers to the cloud and impacting costs. In addition, the WebSocket interface can be used to send messages, such as GUI (Graphical User Interface), to receive and alert you to modified data so that changes in the system are possible to view immediately. Needless to say, the third party's KNX IoT interface provides state-of-the-art IT security.

As a next step, KNX also strives to complete the KNX IoT Point API specification, which is a solution enabling IPv6 communication between sensor/actuator (group). It

allows KNX installations to be extended to devices that use other media such as Thread or WiFi to exchange KNX data. In the first step, group communication would be implemented through pre-configured recipients (brokers). To this end, KNX plans to complete the final vote on the specifications in a year and plans an ETS prototype enabling the configuration of security and group tables in the first prototypes of devices of KNX IoT manufacturers.

### 2.4. Smart Building (Smart Home) Template Design

Essentially, the terms "smart home", "intelligent home", "intelligent building" (IB), and "digital home" mean the same thing. In simple terms, it is a house or flat equipped with modern technologies and appliances that can interact with each other and change depending on certain behaviors (see Figure 12) [40]. The concept of IB can be described and defined in many ways, and often it depends on the area of interest. In his book [40], Bohumír Garlík defines IB as follows: "The intelligent building is a dynamic and responsive architecture, a structurally functional method of structure and building technology that provides productive, economic and environmentally acceptable conditions for each occupant, using a continuous interaction between its four basic elements: the building (material, structure, space), the equipment (automation, controlling, systems), the operation (maintenance, management, operation), and the interrelationships among them."

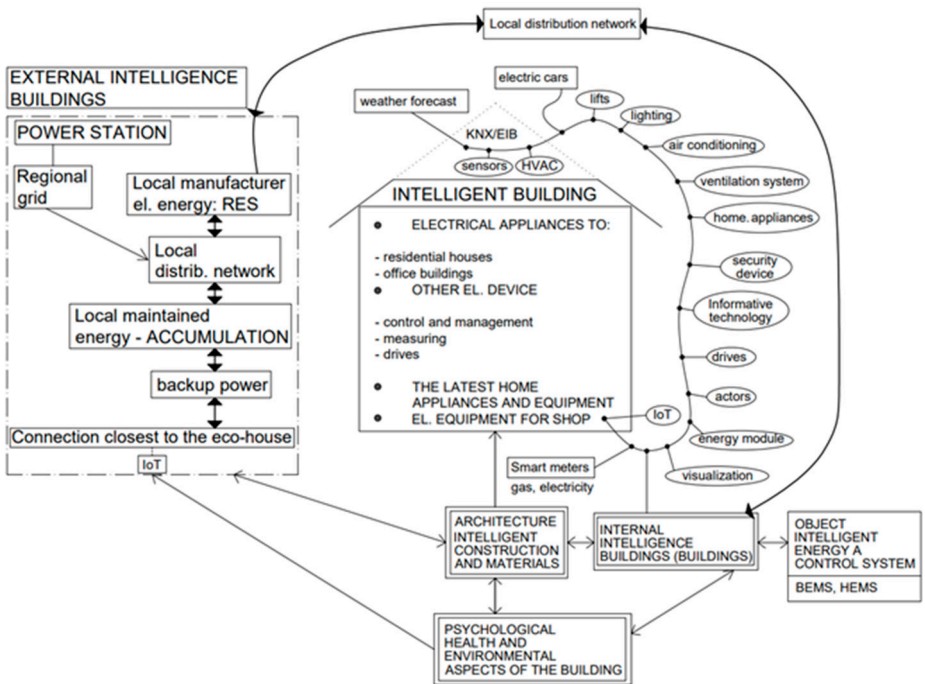

**Figure 12.** Model Process and Structure of an Intelligent Building.

Today's world is a world of technical perfection—perfection in the truest sense of the word, according to the ideas of the Pythagoreans. In the context of intelligent buildings, it is driven by information and communication technologies, energy, smart energy systems (smart grid deployment), IoT, building control systems, robotics, smart materials, sustainability-related technologies, and societal change. In addition to technical development, the built environment will be influenced by many other pressures in the energy and industrial sectors.

Energy consumption impacts cost-effectiveness significantly, and the economic efficiency of housing is the most important aspect of resource conservation. Carbon dioxide emissions produced by fossil-fuel power plants are high, while emissions from renewable sources such as solar energy, wind energy, Earth geothermal energy, biomass, biogas, hydroelectricity, and energy from municipal waste are very low.

IoT Cluster of Buildings Application, Experiment

A household on the first floor within the building cluster with no. 578 was selected. The flat size is 1 + 4 (kitchen and four rooms). The kitchen is superbly equipped with electrical kitchen appliances; there is a photovoltaic system on the building roof, air-conditioning system, and battery storage of electric power, including a charging station in the parking lot of the building cluster. The design of our solution for an automated IB control system based on IoT application is shown in Figure 13. In conjunction with the EPB solution aiming to significantly reduce energy consumption and $CO_2$ emissions, an EMB was designed, where the proposed solution (Figure 13) is implemented in the BEM (see Figure 5).

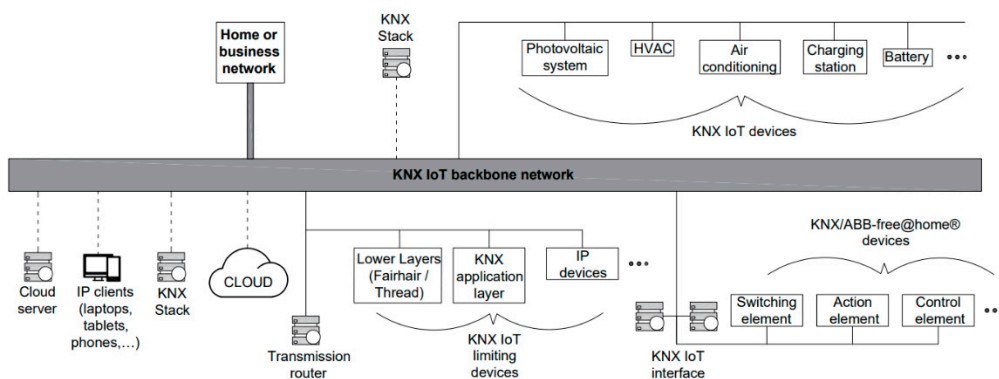

**Figure 13.** Model design of building automation in IoT application.

Figure 13 shows that the new IoT standard of the KNX will provide options in implementing the KNX IoT backbone network via which information is exchanged, and it offers default cloud services.

It is important that the information supplied by the IoT KNX gateways is machine-readable and has rich semantics; ideally, a direct result of the product data used. The rich (semantic) data is simply delivered to the IoT KNX gateway via a specific ETS export, which is more stable than the current XML format and can be upgraded more easily. This enriched data should also offer the possibility to simplify the design phase of a KNX installation (easier selection of supported product functionality), and possibly also to provide a basis for easier interconnection among various products; it should document the installation throughout the lifetime thereof as a part of the building (https://www.knx.org/knx-en/for-professionals/benefits/knx-internet-of-things/index.php, accessed on 4 March 2022).

The whole principle of a smart home is based on the exchange of data and information between individual sensors and devices, which are then used as the basis for further actions. Smart home solutions can be divided according to where various stimuli (data and information) from sensors and devices are processed. There are three possible smart home implementations: local solution, cloud solution, and hybrid solution. For the local solution, which is one application in our case, there is no need to send any information or data via the Internet to a remote server for processing, evaluation, and subsequent sending back to the sensors or devices to perform a certain action. All data is processed and evaluated locally. However, this does not mean that it cannot be controlled remotely when using one of the local solutions. It is possible, however the connection is established directly to the specific device in the household. In most cases, it requires ownership of a public IP address. Here, communication takes place via a bus to connect individual devices. Our case represents a household (flat) that is subject to a decentralized system. Communication among individual devices is peer-to-peer or direct. However, all connected devices must contain their own control unit. The independent functioning of the individual elements guarantees higher reliability of operation [41–43].

The alternative solution used for our flat (smart home) is a cloud solution. It is a solution where all data and information is sent either directly by the devices or through a

gateway via the Internet to a remote server, to a cloud, where it is processed and evaluated. Then, the appropriate commands are sent back via the Internet directly to the actuator—the device—which then takes the corresponding action. We used a cloud solution of ABB Jablonec nad Nisou, Czech Republic, for the IoT application within our smart home.

Cloud services can be controlled using any web browser, and some of them can also be controlled using a mobile phone app. The advantage of using cloud services for smart home management is, for example, the possibility of connecting multiple households (flat and cottage; Smart area, Smart Cities) into one overview and control panel. Furthermore, they can be linked to other Internet services such as weather forecasting, analytics, social media, services for tracking the current location of the household user, and many others. In addition to the advantages of cloud-based home management services, they also bring some disadvantages. For example, higher security risk, the need for a constant fast and stable household connection to the Internet, and more.

### 2.5. KNX Can Be Combined with IoT

The KNX technology is the result of collecting the knowledge and experience gained during the 15 years of its predecessors, which were EIB (European Installation Bus), EHS (European Home System), and BatiBUS.

ISO/IEC is a KNX system approved by the international standard ISO/IEC 14543-3 in 2006. CENELEC is a KNX system approved by the European standard EN 50090 in 2003. CEN is a KNX system approved by the European standard EN 13321-1 (referring to EN 50090) and EN1332-2 (KNXnet/IP) in 2006. SAC is a KNX system approved by Chinese standard GB/Z 20965 in 2007. ANSI/ASHRAE is a KNX system approved by the American standard ANSI/ASHRAE 135 in 2005. Consequently, KNX has become a globally recognized building and household control standard.

The KNX system is currently being connected to a higher-level IoT system. The KNX Association has taken the first step towards standardization on the Internet side by publishing the KNX IoT web services and interface specifications, which allow gateways to be aligned with existing oBIX, OPC/UA web protocols and BACnet web services. It is important that:

- The devices understand each other;
- People can understand what the device or service offers.

Understanding the meaning of data (=semantics) is essential to be able to combine functionality and create new functions. The current KNX system does this by creating a project in ETS, thus relying on standardized KNX data types combining a number of products/functions. The ability to share this semantic information beyond the boundaries of the KNX ecosystem is important for IoT. Semantics allow devices and services to interact at different levels. For example, a wall switch may directly control a particular light fitting, while a mobile phone app may bring the building to "out of home" mode, thereby indirectly controlling the same light fitting. Semantic allows the removal of technical details and, thus, the creation of special values. An example is the command "turn off the heating when the window is open". Another example is "dim the lights in the living room", wherein the distributed system components may decide whether this should result in moving the blinds or dimming the lights. These services work to simplify interaction with building automation systems so that the end-user can control their building space without having to understand the complexities of the building automation system. It means that the defined zones of the given flat can be scheduled automatically and switched on and off according to the user's schedule set directly from the desktop of the computer, tablet, or phone.

In addition to automation systems that use our global Internet, many services have begun to emerge that use the Cloud to support energy savings and transmit important energy, carbon, and financial information to facility staff, green teams, consumers, and financial managers. With these functions and many other options, the energy and comfort conditions can be effectively monitored and controlled to optimize them. In addition, having the almost unlimited storage capacity of the cloud, this data can be securely and

redundantly stored and used throughout the building lifecycle to efficiently monitor and analyze building energy performance, maintenance performance, and cost-saving measures, all at a fraction of the cost associated with embedded systems.

### 2.6. Benefits of a Smart Home (Intelligent Building)

The fundamental advantage that smart homes (intelligent buildings) bring to their users and owners is the long-term reduction in financial costs in the form of energy savings and $CO_2$ reduction. The right combination and usage of smart home technologies will ensure significant energy and cost savings. The possibility of remote control of heating, heating according to the outside temperature or Internet weather forecast can be mentioned as a simple example. Switching the lights depending on the ambient light conditions is another example [44].

Monitoring Energy Consumption in Existing Buildings

When addressing the EPB and certification thereof, the optimal method to monitor energy consumption in buildings is necessary to be chosen. In our case, building no. 578 (Figure 3) will be dealt with, which can be seen in the marked part of the given location of our experiment. First of all, we refer to the CSN 73 0540-2-2011 and the Commission Recommendation (EU) 2016/1318 of 29 July 2016 on guidelines for promoting nearly zero-energy buildings. According to [45], the building is divided into zones, which are defined with respect to the activities taking place within them, and the conditions that are characteristic of each building regarding the activities taking place within them. This standard fully specifies the hourly modelling method for each building zone (flat).

Now, a systematic approach for addressing the EPB will be presented:

1. HVAC solution for the building. The energy balance for heating and cooling takes into account:

    (a) Energy requirements for the heating and cooling needs of each zone in the building/flat.
    (b) The energy supplied to the HVAC system from the RES system.
    (c) Input energy to the HVAC system.
    (d) Energy losses passing through the building envelope.
    (e) HVAC system operation (production, storage, distribution, energy transmission).

2. Solution of the internal environment of the building (flat):

    (a) Distributing heat within a room and achieving the conversion of indoor conditions to the desired heating or cooling, taking into account energy consumption.
    (b) Adjustment of temperature fluctuations (vertical fluctuations in room temperature increase the average air temperature causing thus heat losses in rooms with a large height between the lower and upper floors).

3. The HVAC systems can reduce heat loss depending on the height between the floor and ceiling (room height), affecting the energy consumption for their operation.

### 2.7. Objective Function

The objective evaluation function is based on the prices of individual energy media entering the EnergyHub (central technical room, where there are: a boiler for heating, central air-conditioning unit, cogeneration, photovoltaic system distributor, central water supply, sewage water, etc.) and the daily consumption thereof. The criterion for evaluating this function is the sum of the costs for these energy types.

$$min\ OF = \sum_t \lambda_t^a A_t + \lambda_t^b B_t + \ldots + \lambda_t^n N_t \tag{1}$$

where $\lambda_t^a\ \ldots\ \lambda_t^n$ are the unit prices of the input commodities, and $A_t \ldots N_t$ are the consumptions of these commodities.

### 2.7.1. Demand for Energy Consumption

The amounts of energies that enter the EnergyHub and are needed to satisfy demand through the conversion, production or storage of these energies represent the components of the EnergyHub, given by the sum of the energies entering the EnergyHub components [46].

$$P(t) = \sum_t A_{i,t} + A_{j,t} \ldots + A_{n,t} \tag{2}$$

where $A_{i,t} \ldots A_{n,t}$ are the amounts of energy input to the EnergyHub.

The amounts of energy that output from the Energy Hub and are needed to satisfy demand are given by the sum of the energies outputting from the Energy Hub components [47].

$$L(t) = \sum_t B_{i,t} + B_{j,t} \ldots + B_{n,t} \tag{3}$$

where $B_{i,t} \ldots B_{n,t}$ are the amounts of energy output from the EnergyHub.

### 2.7.2. Energy Conversion Efficiency

When implementing the process of technical energy conversion from one form to another, there are limitations on the energy flow density at the output of the conversion system. Each technical component has a certain limit based on the size of the given component and the material's ability to withstand temperature, revolutions, and electric current. The device cannot exceed this limit. The conversion, conversion efficiency, and operating ranges are expressed as [47]:

$$B_t^{\min} < B_t < B_t^{\max} \tag{4}$$

$$B_t = \eta_x A_t \tag{5}$$

$$\eta_x < 1 \tag{6}$$

where $B_t^{min}$ is the minimum operational performance of the component, $B_t$ is the instantaneous operational performance of the component, and $B_t^{max}$ is the maximum operational performance of the component. The component functioning can be expressed as energy production $B_t$ by consumption of an input commodity $A_t$ with a given efficiency $\eta_x$. Primary efficiency $\eta_x$ is always less than 1 [47].

### 2.7.3. Energy Storage Limitation

The EnergyHub can have three possible types of storage–cold, heat, and electric power storage. In principle, however, all of these storage types work in the same way, which can be expressed by the following equations [48]:

$$SOC_t = SOC_{t-1} + \left( A_t \eta_a - \frac{B_t}{\eta_b} \right) \Delta t \tag{7}$$

$$A_{min} \leq A_t \leq A_{max} \tag{8}$$

$$B_{min} \leq B_t \leq B_{max} \tag{9}$$

$$SOC_{min} \leq SOC_t \leq SOC_{max} \tag{10}$$

Variable $SOC$ represents the storage state of the charge, $A_t$ is the amount of energy inputting to the storage, and $B_t$ is the amount of energy outputting of the storage. The first Equation (7) shows the change in charge level from the previous state with the efficiencies of charging $\eta_a$ or discharging $\eta_b$. The limits for charging or discharging the storage are limited by maximum and minimum values, i.e., $A_{min}$, $A_{max}$, $B_{min}$, $B_{max}$, and the storage charge level is defined by the minimum storage capacity value $SOC_{min}$, while the maximum storage capacity value $SOC_{max}$ [48].

### 2.7.4. Transformer

The transformer is a non-rotating electrical device allowing the conversion of alternating voltage. It consists of two windings. For the purposes of the EnergyHub mathematical model, the transformer can be expressed as follows [49]:

$$E_t^{out} = \eta_{ee} E_t^{in} \tag{11}$$

The transformation of the electric voltage level is not lossless, so the transformer efficiency also needs to be introduced $\eta_{ee}$ [50].

### 2.7.5. Cogeneration Unit

The cogeneration unit is a device for combined electrical power and heat production. Natural gas is the most commonly used fuel for cogeneration units [51]. Mathematically, the combined production of electrical power and heat can be described as follows:

$$H_t = \eta_{gh}^{chp} G_t \tag{12}$$

$$E_t = \eta_{ge}^{chp} G_t \tag{13}$$

The first Equation (12) expresses the amount of heat $H_t$ produced by the cogeneration unit through the combustion of gas $G_t$ with an efficiency $\eta_{gh}^{chp}$. The second Equation (13) then expresses the amount of electrical power $E_t$ produced by combustion of a quantity of gas $G_t$ with an efficiency $\eta_{gh}^{chp}$ [52].

### 2.7.6. Boiler

The boiler is a simple combustion device where heat is generated by burning fuel. The initial mathematical model uses natural gas as fuel [53]. This simple process of producing heat energy by gas combustion in a boiler is expressed by the following equation:

$$H_t = \eta_{gh} G_t \tag{14}$$

where $H_t$ is the amount of thermal energy produced, $\eta_{gh}$ is the boiler efficiency, and $G_t$ is the amount of fuel [49].

### 2.7.7. Heat Pump

A heat pump is a device that pumps heat from a colder place to a warmer place, exerting external work. Electrical energy is supplied to the heat pump inlet to power the compressor. It compresses the refrigerant gas passing to the condenser, where it releases its latent heat and heats the heating substance (e.g., heating water). Subsequently, the refrigerant passes through the pressure reducing valve to the evaporator, where it receives the latent heat at a lower pressure and temperature, thus cooling the refrigerant circulating through the evaporator (e.g., cooling water), and passes back to the compressor, and the whole process is repeated [54]. The following equations mathematically express the whole process of heat pump operation:

$$C_t + H_t = E_t \times COP \tag{15}$$

$$H_t^{min} I_t^h \leq H_t \leq H_t^{max} I_t^h \tag{16}$$

$$C_t^{min} I_t^c \leq C_t \leq C_t^{max} I_t^c \tag{17}$$

$$I_t^c + I_t^h \leq 1 \tag{18}$$

$$I_t^c, I_t^h \in \{0,1\} \tag{19}$$

The heat pump can operate in either cooling or heating mode. These states are expressed by the variables $I_t^c$ or $I_t^h$. Amount of cooling capacity $C_t$ or heating capacity $H_t$ is

given within the limits of the minimum cooling capacity $C_t^{min}$ or heating capacity $H_t^{min}$ and the maximum cooling capacity $C_t^{max}$ or heating capacity $H_t^{max}$ [49].

### 2.7.8. Energy Price

For calculating the objective evaluation function, the determination of the energy price is the critical factor. The energy price of each commodity can be specified as dynamic depending on a time variable–such as different price tariffs for electric power purchase. Another option is a fixed price that remains constant throughout the day–this is the case for the price of natural gas.

### 2.7.9. Heat Consumption for Heating

The hourly heat consumption for heating is determined according to the relation: [49]

$$Q_{VYT,h} = Q_c \tag{20}$$

where $Q_{VYT,h}$ is the hourly heat consumption for heating and $Q_C$ is the heat loss of the building. The heat loss of the building can be expressed as [49]

$$Q_c = U_{em} \times S \times (t_i - t_e) \tag{21}$$

where $U_{em}$ is the mean heat transfer coefficient, $S$ is the building envelope area, and $(t_i - t_e)$ is the difference between the exterior and interior temperature values.

The exterior design temperature for Prague has been set at $-12\,°C$, and the interior temperature has been set at $20\,°C$.

The hourly heat consumption for heating is determined from building parameters and temperatures and is summarized in Table 3 [49].

Overall result: Hourly heat consumption for heating in the area is: 2367.25 kWh.

### 2.7.10. Heat Consumption for Hot Water Heating

The hourly heat consumption for hot water heating is determined based on the daily heat consumption for hot water heating, which is determined according to the following equation

$$Q_{TV,d} = \frac{\rho \times c \times V_{2p} \times (t_{TV} - t_{SV})}{3600} \tag{22}$$

where, for water, the density is $\rho = 1000\,kg/m^3$, the specific heat capacity $c = 4.182\,kJ/kgK$, $V_{2p}$ is the total hot water demand for the consumption of all persons ($0.082\,m^3/person/day$ for those living in the block of flats), $t_{TV}$ is the hot water temperature ($55\,°C$), and $t_{SV}$ is the cold water temperature ($10\,°C$) [55].

Hourly HW demand is the daily HW demand per hour, i.e., divided by the number of hours in the day, i.e., 24 h.

$$Q_{TV,h} = \frac{Q_{TV,d}}{\tau} \tag{23}$$

where $Q_{TV,h}$ is the hourly heat consumption for hot water heating, $Q_{TV,d}$ is the daily heat consumption for hot water heating, and $\tau$ is the time period (24 h in the case of calculating hourly consumption from daily consumption).

**Table 3.** Hourly Heat Consumption for Heating After Reconstruction.

| Building Category | Building Envelope Area | Mean Heat Transfer Coefficient | Exterior Temperature | Interior Temperature | Heat Loss of the Building | Hourly Heat Consumption for Heating | Number of Buildings under the Given Category | Hourly Heat Consumption for Heating under the Given Category |
|---|---|---|---|---|---|---|---|---|
| | $S$ [m$^2$] | $U_j$ [Wm$^2$/K] | $t_e$ [°C] | $t_i$ [°C] | $Q_c$ [W] | $Q_{VYT,h}$ [kW] | | $Q_{VYT,h}$ [kW] |
| Family house no. 576 | 1352 | 0.292 | −12 | 20 | 12,633.08 | 12.633 | 61 | 770.61 |
| Family house no. 577 | 950.2 | 0.262 | −12 | 20 | 8492.99 | 8.492 | 47 | 399.12 |
| Block of flats no. 578 | 1814.52 | 0.275 | −12 | 20 | 15,967.77 | 15.967 | 75 | 1197.52 |
| | | | | | | | Total [kW] | 2367.25 |

### 2.7.11. Gas Consumption

The prerequisite for determining the gas consumption in a given area is that the gas is only used for cooking. The annual average cooking gas consumption is estimated at 200 kWh/person/year. Since this is a given amount of gas consumption per person per year, and the price of gas is usually constant throughout the year, this annual gas consumption is budgeted equally for each hour regardless of the hour of the day:

$$Q_{p,d} = \frac{Q_{p,r}}{\tau} \tag{24}$$

where $Q_{p,d}$ is the daily gas consumption, $Q_{p,r}$ is the annual gas consumption, and $\tau$ is the number of days in a year (365 or 366 days in the case of a leap year).

$$Q_{p,h} = \frac{Q_{p,d}}{\tau} \tag{25}$$

where $Q_{p,h}$ is the hourly gas consumption, $Q_{p,d}$ is the daily gas consumption, and $\tau$ is the number of hours in a day (24 h).

$$Q_{p,d} = \frac{200}{365} \tag{26}$$

$$Q_{p,d} = 0.55 \text{ kW/person per day} \tag{27}$$

$$Q_{p,h} = \frac{0.55}{24} \tag{28}$$

$$Q_{p,h} = 0.02 \text{ kW/person per hour} \tag{29}$$

The hourly cooking gas consumption per person is calculated at 0.02 kW.

Cooking gas consumption is summarized in the table in the "Results" section.

### 2.7.12. Electric Power Consumption

The electric power consumption per hour per household can be read from the recalculated Type Delivery Diagrams (TDDs), retrospectively for the past period. TDD can be used to replace continuous metering for customers with non-continuous metering of type C. The application of TDD is described in Decree 541/2005 Coll., on electricity market rules. TDDs are used to estimate the likely consumption pattern of a specific consumption point over time and take into account a variety of factors that influence consumption modelling [56].

Specifically, TDD No. 4, which is designed for households without thermal usage of electric power, will be used. Type delivery diagrams were obtained from the OTE website [57]. Their calculation is as follows:

$$O_h = O_r * \frac{r_h}{\sum_{h=1}^{8760} r_h} \tag{30}$$

where $O_h$ is the hourly electric power consumption, $Or$ is the annual consumption in the consumption point, $r$ is the hourly coefficient, and $\sum_{h=1}^{8760} r_h$ is the sum of the hourly coefficients of the whole given year [49].

### 2.8. RES in Conservation Areas

Buildings under protection and buildings perceived to be of cultural value are specific to the given area and historical period in terms of form. They are legally protected from changes related to their location (urban planning), materials, technologies used during construction (historic structures), and visual look (roofscape).

Therefore, for the RES design, it is necessary to anticipate not only the lifetime of the equipment or the return on investment but also the potential impacts on the surrounding development and the character of the building and locality. Then, the criteria for the RES integration are mainly related to the aesthetic appearance and possibilities of the territory. Solar energy has great potential.

For applications in buildings with a cultural value, an integrated installation that fully replaces the structure (e.g., BIPV as roofing and, at the same time, PV) does not damage the structure of the building and does not distort the external shape and structure of the construction is an ideal case. In foreign publications, visibility assessment from starting points (e.g., public spaces, scenic points, panoramas, etc.) is mentioned. BIPV includes all photovoltaic application technologies, particularly thin-film technologies such as CdTe, CIGSe, a-Si te- (a-Si/μc-Si), and organic PV technologies [58].

Integrated panels are available in more colours than conventional photovoltaic panels. The panel coloring is problematic in terms of color fastness and application to the layers, but it provides the desirable variety. Coloring affects the reflectance of sunlight and, therefore, the overall performance of the BIPV. Examples of colored BIPV include monocrystalline solar panels in the HJT (HIT) or PERC production variant made by SOLAXESS. Conventional monocrystalline panels are complemented by a colored layer and a top protective ETFE film. In addition, the coloring inevitably affects their performance (see Figure 14) [59].

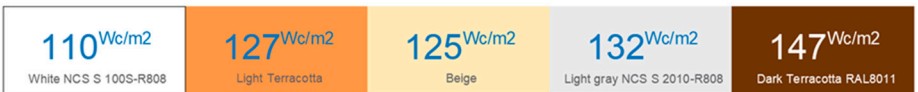

**Figure 14.** Solar panel performance and color.

In the Czech Republic, systems of (non-)connection to a power grid are used generally:

- Off-grid–so-called island PV systems. The inverter does not supply power to the grid; it is isolated from the public grid. Consumption of self-generated energy is preferred. In case of a power shortage and connection to the grid, it is possible to switch to mains consumption and recharge the batteries. When wiring, appropriate appliances operating under DC current or having a voltage converter (inverter) installed are advisable to choose [59].
- On-grid. The system is connected to the grid and supplies electricity to the public grid.
- Hybrid. This allows, for example, a zero overflow setting and energy storage in batteries.

### 2.9. Power Control

Bypass diodes are advisable to be placed at the level of the photovoltaic panels. Some modules have diodes incorporated in their structure. The diodes switch to the forward direction at a negative voltage, which is caused by the shading of the solar cell equal to the sum of the unshaded cell voltage and the bypass diode voltage.

The photovoltaic electricity generation is associated with fluctuations caused by climatic conditions, such as cloud shading. Ideally, the operation works at the MPPT point. It can be achieved in several ways:

- Control on the panel side (by switching their serial or parallel arrangement);
- On the heating resistance side (by changing the internal resistance);
- Using the controller included between the panels and the heating resistance.

The first two options usually provide only a few operation combinations, thus allowing only a partial approach to MPP. The use of mechanical contactors is not very suitable in terms of switching frequency; the wiring using a series of semiconductor switches is not suitable in terms of their cost and the need for many pieces with serial sequencing [60]. Another control option for consumer connection is using a three-phase inverter that allows asymmetrical control or connecting three single-phase inverters.

### 2.10. Electricity Storage

Storage for hot water is suitable for smaller hot water consumption or a more cost-effective option for buying energy supplied to the grid at a low price.

Battery storage is suitable wherever overproduction of electric power is expected. Given the cost of buying electric power, the energy is paid off to be stored most of the time.

*2.11. Municipal Energy Centre (MEC)*

The Municipal Energy Centre responds to sustainable energy use (sustainable power engineering) and reduces $CO_2$ emissions. It is a decentralized source of all energy and technical systems connected to a specific location. They include additional equipment that the space of the buildings themselves would not be able to accommodate. MEC focuses on obtaining energy from renewable and alternative energy sources. The aim is to provide a stable system in the locality that will cover the energy supply to the buildings of the locality and maximize RES usage [60] This is the fundamental purpose of this research work. The IoT plays an indispensable role in this context. It is actually at the heart of the solution regarding regulatory measures towards energy self-sufficiency and, in essence, energy sustainability. The MEC implementation is functionally embedded in the Building Energy Model, integrating IoT, Cloud, and building control systems (Figure 5).

Design of the MEC Mathematical Model as an EnergyHub

The mathematical model of the EnergyHub for a given variant is defined in Figure 15 by the following equations:

$$min\ OF = \sum_t \lambda_t^e E_t + \lambda_t^g G_t \tag{31}$$

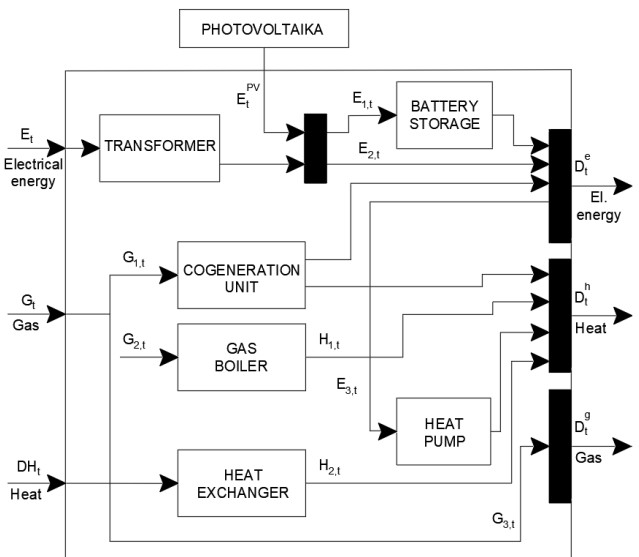

**Figure 15.** Mathematical Model of EnergyHub.

The objective evaluation function of the mathematical model *OF* is given by the sum of electric power prices $\lambda_t^e$, gas prices $\lambda_t^g$ and electric power consumption $E_t$ and gas consumption $G_t$. By minimizing it, the optimal operation of the EnergyHub is obtained to minimize the energy cost under the given conditions [46].

Let us express the equation of the electric power flow to the EnergyHub. Here, there is the expression for the input electric power flow from the power grid $E_t$ through a transformer of efficiency $\eta_{ee}$ to the EnergyHub, where it is expressed as $E_{1,t}$ [46]:

$$\eta_{ee} E_t = E_{1,t} \tag{32}$$

Next, let us express the equation of the electric power flow from the EnergyHub. The left side of the equation consists of the electric power from the grid $E_{1,t}$ and the electric power generated by a cogeneration unit $\eta_{ge} G_{1,t}$ (production from the gas amount of $G_{1,t}$ with an efficiency $\eta_{ge}$); the right side of the equation consists of the demand for electric

power by consumers in the given locality $D_t^e$ and the electric power required to operate the heat pump $E_{2,t}$ [46]:

$$E_{1,t} + \eta_{ge}G_{1,t} = D_t^e + E_{2,t} \tag{33}$$

Now, let us express the gas flow equation to the EnergyHub. Gas flow at the input to the EnergyHub $G_t$ is divided into the gas flow for the gas boiler $G_{2,t}$, the gas flow for the cogeneration unit $G_{1,t}$, and the gas flow for direct consumption by users $G_{3,t}$ [46]:

$$G_t = G_{1,t} + G_{2,t} + G_{3,t} \tag{34}$$

Now, let us express the equation of the gas flow from the EnergyHub. The equation assigns the input portion of the gas flow $G_{3,t}$ to the consumer's gas demand for direct consumption $D_t^g$ [46]:

$$G_{3,t} = D_t^g \tag{35}$$

Next, let us express the equation that expresses the heat energy output from the EnergyHub. The left side of the equation consists of the thermal energy produced by the cogeneration unit (production from the gas amount of $G_{1,t}$ with an efficiency $\eta_{gh}$), the thermal energy produced by burning gas in the gas boiler $H_{1,t}$ and the thermal energy produced by the heat pump $H_t^{EHP}$; the right side of the equation consists of the demand for thermal energy by consumers connected to the EnergyHub in the specified locality $D_t^h$ [46]:

$$\eta_{gh}G_{1,t} + H_{1,t} + H_t^{EHP} = D_t^h \tag{36}$$

The heat combustion in the gas boiler is described by the following equation, where the left side represents the gas combustion efficiency in the gas boiler $\eta_{gh}^f$ and the gas amount gas $G_{2,t}$ required to produce heat energy $H_{1,t}$:

$$\eta_{gh}^f G_{2,t} = H_{1,t} \tag{37}$$

The equations below describe the heat pump's operation in heat production mode only. The first two equations express the operating state of the heat pump $I_t^h$; the third equation expresses the performance of the heat pump $H_t^{EHP}$, which is given by the heating factor of the heat pump $COP$ and the input power of the heat pump $E_{2,t}$; and the fourth equation expresses the operational performance limits of the heat pump, where $H_t^{min}$ and $H_t^{max}$ are the minimum and maximum performance of the heat pump, respectively.

$$I_t^h \leq 1 \tag{38}$$

$$I_t^h \in 0, 1 \tag{39}$$

$$H_t^{EHP} \leq E_{2,t} \times COP \tag{40}$$

$$H_t^{min} I_t^h \leq H_t^{EHP} \leq H_t^{max} I_t^h \tag{41}$$

In the variant numbered 3, which was chosen in relation to the given urban area, battery storage is included in the model. The following equations express the final mathematical expression of the EnergyHub.

Photovoltaic panels are added to the system. For calculation purposes, the electric power flow equations at the input to EnergyHub is modified:

$$\eta_{ee}E_t + E_t^{PV} = E_{1,t} + E_{2,t} \tag{42}$$

In Equation (42) for the electric power entering the EnergyHub, a term $E_t^{PV}$ expressing the electric power obtained from the photovoltaic system is introduced [46]. In Equation (42) of the electric power flow to the EnergyHub, a term $E_{1,t}$ determining the electric power flow for the battery storage and a term $E_{2,t}$ determining the electric power flow for direct

consumption of electric power by consumers connected to the EnergyHub are introduced on the equation right side.

Next, a term $E_t^{dch}$ of the electric power flow to the battery storage is added to the equation of power flow from the EnergyHub, and the indices of the other electric-power flow terms are modified:

$$E_{2,t} + E_t^{dch} + \eta_{ge} G_{1,t} = D_t^e + E_{3,t} \tag{43}$$

An equation is added to the model to assign the flow of electric power $E_{1,t}$ as the flow of electric power to the battery storage $E_t^{ch}$. The equations below define the operation of battery storage for electric power. The first Equation (44) expresses the change in battery storage charge, where $SOC$ is the state of charge of the battery storage, $E_t^{ch}$ is the electric power flow for charging the battery storage with charging efficiency $\eta_c$, and $E_t^{dch}$ is the electric power flow from the battery storage with discharge efficiency $\eta_d$. The second Equation (45) expresses the operating state of the battery storage, where $I_t^{dch}$ expresses the discharge and $I_t^{ch}$ expresses the charging of the battery storage. The third (46) and fourth (47) equations express the charging and discharging limits of the battery storage. The minimum and maximum charging or discharging limits are expressed by the term $E_{min}^{ch}$ for minimum charging, $E_{max}^{ch}$ for a maximum charge, $E_{min}^{dch}$ for minimum discharge, and $E_{max}^{dch}$ for maximum discharge. The last Equation (48) expresses the battery storage capacity. Battery storage charge status (49) $SOC_t$ has to be within the minimum state of charge $SOC_{min}$ and the maximum state of charge $SOC_{max}$ [46]:

$$SOC_t = SOC_{t-1} + \left( E_t^{ch} \eta_c - \frac{E_t^{dch}}{\eta_d} \right) \Delta t \tag{44}$$

$$I_t^{dch} + I_t^{ch} \leq 1 \tag{45}$$

$$I_t^{ch}, I_t^{dch} \in 0, 1 \tag{46}$$

$$E_{min}^{ch} I_t^{ch} \leq E_t^{ch} \leq E_{max}^{ch} I_t^{ch} \tag{47}$$

$$E_{min}^{dch} I_t^{dch} \leq E_t^{dch} \leq E_{max}^{dch} I_t^{dch} \tag{48}$$

$$SOC_{min} \leq SOC_t \leq SOC_{max} \tag{49}$$

The last change made to the mathematical model for Option 3 is a change in the electric power flow index for the heat pump.

## 3. Results

Figure 3 shows the locality of our experiment where we addressed the EPB issue through a novel approach, i.e., using BEM along with an automated building control system on the KNX/FOXTROT/ABB-free@home® platform, see Figure 4. The use of the IoT platform was our main goal. For this purpose, a new EPB model has been developed, namely the BEM with the integration of IoT, Cloud and building control systems, which is presented and described in Figure 5.

The result of our EPB research solution is the reduction of energy consumption and $CO_2$ production on the IB platform, where the model and structure of IB are shown in Figure 12.

Our goal was defined in the introductory chapter of this paper and was divided into four areas:

1. Perform analysis of IoT applied protocols and transmission systems.
2. Define and design the basic energy model of the building(s) with the IoT integration.
3. Design the smart building (intelligent building (IB), smart home, smart office, etc.) template.

4.  Solve the EPB issue of the urban area of Prague 6–Bubeneč, Vítězné nám., when applying BEM with the integration of IoT, cloud, and automated building control systems (Figure 5).

*3.1. Solutions and Results of Goal Fulfilling*

- Point 1 of our goal has been fulfilled, and the solution thereof is presented in Chapter 2.1, wherein Chapter 2.2 describes the structure of the IoT standards and protocols used, based on the analysis of available protocols. The respective protocols were compared, taking into account their effective application from all possible approaches that are required by IoT usage.
- Point 2 of our goal was to define and design the basic BEM (building) with IoT integration. This point has been fulfilled, wherein this initial model has been designed (see Figure 5) with a specific description of the function thereof.
- Point 3 of our goal has been fulfilled, wherein the smart building (intelligent building) template has been designed. Figure 13 describes the new IoT standard of KNX association in the process of the application thereof on the smart home platform.

A basic prerequisite for fulfilling this goal was to monitor the energy consumption of the existing buildings within our experiment. This section discusses the optimal way to monitor energy consumption in existing buildings in order to renovate them to meet the NZEB requirement. This remote and automated approach can be used for building monitoring and can also contribute to building certification, including the design of specific building parameters and design composition, including smart control systems for technical building installations (TBI). There are two issues related to the indoor environment of a building and associated with the heat distribution within rooms and achieving the conversion of indoor conditions to the desired heating or cooling, taking into account energy consumption. To this end, Chapter 2.1addresses the issue of monitoring energy consumption in existing buildings. Subsequently, Chapter 2.8 deals with the RES application in conservation areas and, finally, a municipal energy center (MEC) in the specified area of Prague 6–Bubeneč is defined and applied in Chapter 2.11. The MEC application is a conclusive basis to address energy sustainability and energy efficiency along with a significant reduction of energy consumption in buildings while substantially reducing $CO_2$ production. It is a decentralized source of all energy and technical systems connected to a specific location. It includes additional equipment that the space of the buildings themselves would not be able to accommodate. MEC focuses on obtaining energy from renewable and alternative energy sources. The aim is to provide a stable system in the locality that will cover the energy supply to the buildings of the locality and maximize RES usage.

The fourth goal of our research was to validate the methodological procedure of solving EPB through the new EMB according to Figure 5, together with the application of the EPB solving principle according to Figure 6 within our experiment.

*3.2. Evaluation and Design of EPB within the Experiment*

3.2.1. Evaluation of the EPB Is Based On

- Directive 2012/27/EU of the European Parliament and of the Council of 25 October 2012 on energy efficiency, amending Directives 2009/125/EC and 2010/30/EU and repealing Directives 2004/8/EC and 2006/32/EC as amended by the 2018 revision of this Directive (hereinafter referred to as "Directive 2012/27/EU").
- Act No. 406/2000 Coll., on energy management, as amended (hereinafter referred to as "Act No. 406/2000 Coll."). The State Energy Concept is adopted for a period of 25 years and is binding for the state administration in the field of energy management.

3.2.2. Greenhouse Gas Emissions

The EU's 2030 Climate and Energy Policy Framework has set an EU-level target to achieve a reduction of at least 40% in greenhouse gas emissions by 2030 compared to



1990. This target breaks down further into targets for emission reductions of 43% and 30% compared to 2005 levels in the EU ETS sectors and non-EU ETS sectors, respectively.

The targets for individual Member States range from 0 to 40% compared to 2005. For the Czech Republic, the Regulation sets a binding target of a 14% reduction in emissions compared to 2005 and a binding linear trajectory to achieve it, starting at the average of greenhouse gas emissions for 2016, 2017, and 2018, and ending in 2030.

### 3.2.3. Renewable Energy Source Share

The Czech Republic plans to achieve a share of renewable energy in the final gross consumption of 22% by 2030, an increase of 9 percentage points compared to the national target of 13.0% for 2020. The said share of 22% corresponds to the requirement to express a national contribution to the EU binding target of 32.0% by 2030 under Article 3 of the revised version of Directive 2009/28/EC on the promotion of the use of energy from renewable sources (Directive 2018/2001).

### 3.2.4. Energy Efficiency Dimension (Energy Consumption)

Directive 2012/27/EU (Art. 3) allows each Member State to set an indicative national contribution to the EU target based on primary energy consumption or final energy consumption, primary energy savings or savings in final energy consumption, or energy intensity. At the same time, however, Member States are to respect the EU's energy efficiency target for 2020 and 2030, which is set at 20% and 32.5%, respectively. The implementation of this target should result in EU primary energy consumption of no more than 1 474 Mtoe or final energy consumption of no more than 1 078 Mtoe in 2020.

#### Solving EPB with DesignBuilder Simulation

Prior to the proposed reconstruction of the building cluster, of the locality of Vítězné nám, Prague 6 (experiment), a simulation of the buildings in the named location, was carried out using DesignBuilder software for later comparison of savings in energy consumption and reduction in $CO_2$ production. The savings' demonstration value will be the difference of the energy consumption indicators in kWh/(m$^2$.y); see Tables 4 and 5.

**Table 4.** Energy Consumption Values in kWh/(m$^2$.y) Before Reconstruction.

| Building | Energy Supplied | Total Energy |
|---|---|---|
| No. 576 | 85.22 | 207.32 |
| No. 577 | 91.25 | 177.32 |
| No. 578 | 133.76 | 233.34 |

**Table 5.** Energy Consumption Values in kWh/(m$^2$.y) After Reconstruction.

| Building | Energy Supplied | Total Energy |
|---|---|---|
| No. 576 | 85.13 | 207.2 |
| No. 577 | 74.16 | 152.17 |
| No. 578 | 107.22 | 205.62 |

Evaluation of the simulation results of the building cluster reconstruction: According to Table 1, a value of supplied energy of 50 to 70 kWh/(m$^2$.y) was required. Tables 4 and 5 show that the energy consumption after building cluster reconstruction decreased by 12.6% on average. The pattern of consumption is shown in Figure 16.

### 3.2.5. Internal Insulation

The building envelope will be insulated according to the requirements of CSN 73 0540-2:2011 for the heat transfer coefficient $U_{N,20}$ = 0.3 W/(m$^2$.K).

The 75-cm thick brickwork wall will be insulated with a 100-mm Multipor board.

The 60-cm thick brickwork wall will be insulated with a 125-mm Multipor board.

The 45-cm thick brickwork wall will be insulated with a 125-mm Multipor board.

The 30-cm thick brickwork wall will be insulated with a 125-mm Multipor board.

Note: External insulation cannot be carried out because the buildings are listed buildings, and the reconstruction of the building envelope has not been approved by the Heritage Authority. See Figure 17 for an example of the building. Only the heat transfer coefficient changes due to the reconstruction of the inner walls and windows. The heat transfer coefficients before and after the reconstruction are given in Table 6.

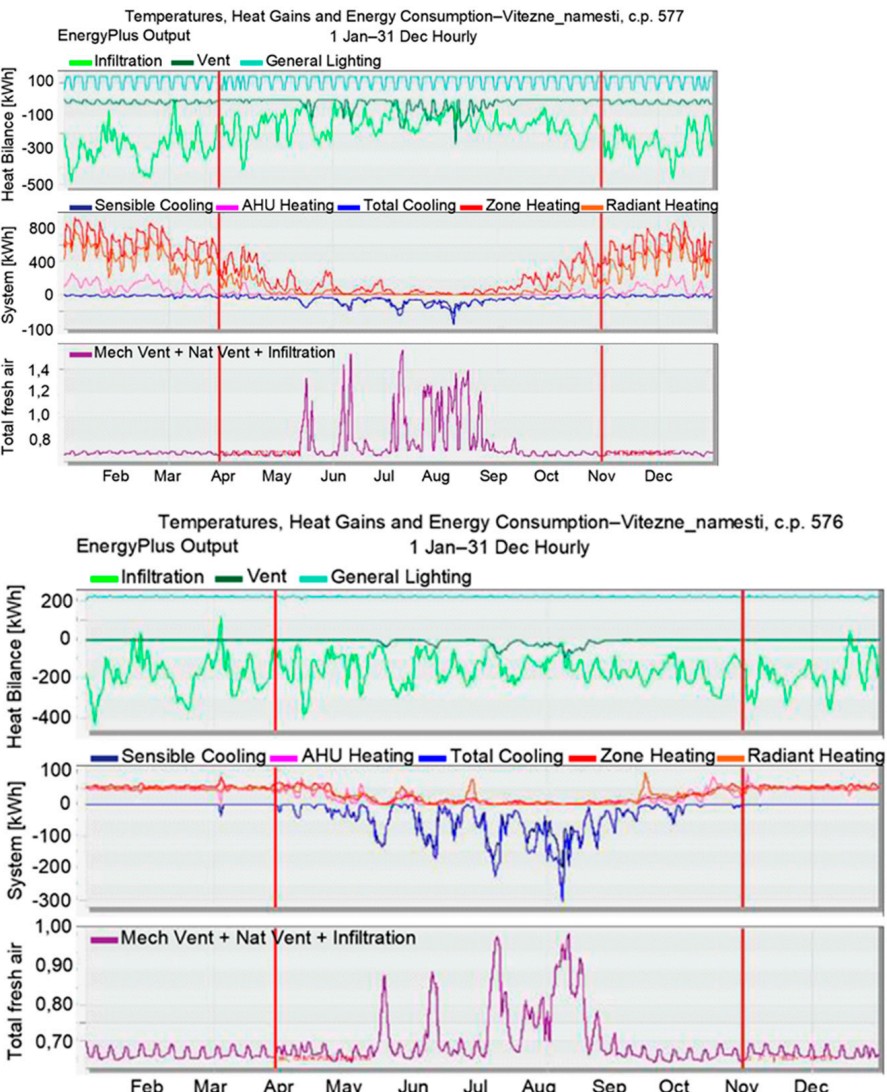

**Figure 16.** The Pattern of Energy Consumption.

**Table 6.** Heat Transfer Coefficients Before and After Reconstruction (Insulation).

| Structure | Heat Transfer Coefficient before Reconstruction U [W/(m².K)] | Heat Transfer Coefficient after Reconstruction U [W/(m².K)] |
|---|---|---|
| Wall thickness of 75 cm | 0.872 | 0.292 |
| Wall thickness of 60 cm | 1.033 | 0.262 |
| Wall thickness of 45 cm | 1.267 | 0.275 |
| Wall thickness of 30 cm | 1.62 | 0.289 |

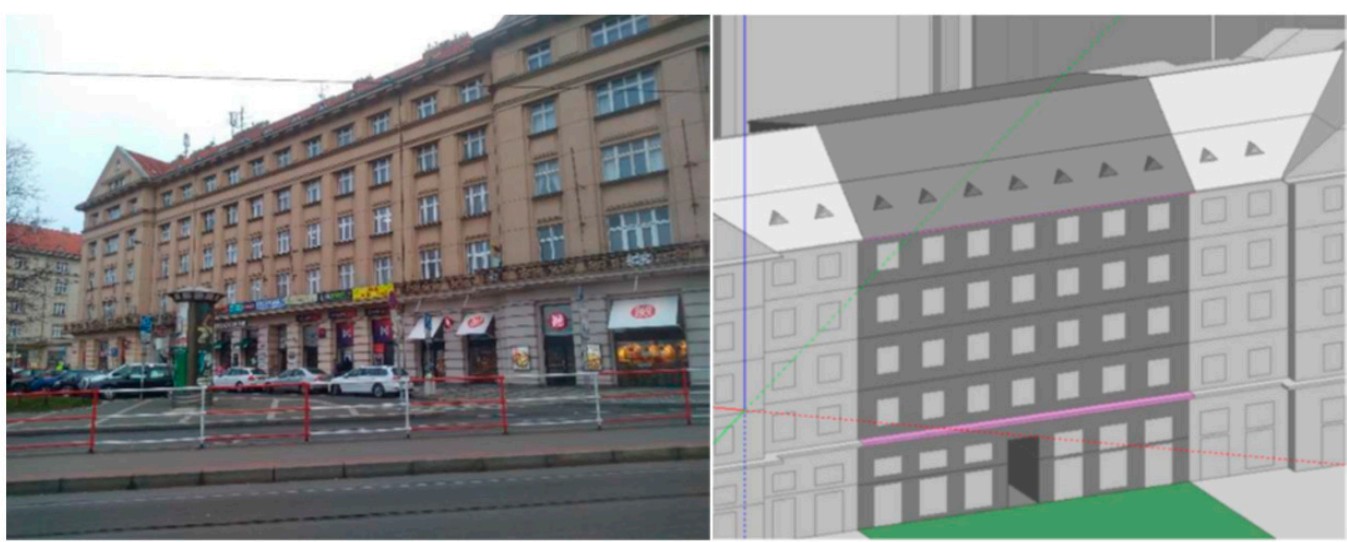

**Figure 17.** Block of Flats with Parterre. Vítězné náměstí 577/2, Dejvice, Prague 6, Czech Republic.

### 3.2.6. Glazing

The estimated heat transfer coefficient with the original glazing is 2.42 W/(m$^2$.K). In order to meet the requirements, the external glazing will be replaced with double glazing 4-8-4 with heat transfer U = 1.4 W/m$^2$.K (CSN 730532). The internal glazing will be retained if it is found to be in good condition. When replacing the interior glass as well, a simple FLOAT 4 mm (clear) glass will be installed. As a result of the insulation, energy costs are likely to be reduced, especially for heating. For comparison, the following table shows the energy consumption before and after the reconstruction.

### 3.2.7. Risk of Condensation

The preliminary design, which stated the possibility of water condensation in the structure, was refuted after further simulation in Wufi. No condensation occurs in the structure during the year.

### 3.3. *The Influence of the Application of Automated Control Systems in the Context of IoT*

In order to achieve a 20% or 32.5% reduction in energy consumption, it would be necessary to focus on automated control systems for individual households (flats) in the given localiy. Referring to the 2018 study of the "Central Association of the Electrical and Electronics Industry" (ZVEI–the main German trade association: Zentralverband Elektrotechnik–und lektronikindustrie e. V.), it states: Conclusion of the research on the issue of reducing energy consumption in the application of KNX automated control system: The use of modern electrical installation systems represents a significant potential for reducing energy consumption. Overall, when measures are implemented to optimise (technologies) control, the average energy savings range from 11 to 31%. When carrying out a reconstruction proposal for our urban area (cluster of buildings), energy savings of 12.6% were achieved. In the case of implementation of automation using KNX or FOXTROT control systems, or an ABB-free@home® system, including IoT applications, energy consumption will be reduced by 11 to 31%. Finally, a 23.6 to 43.6% reduction in energy consumption in the given renovated locality can be achieved. It means that the energy reduction requirement according to Directive 2012/27/EU (Article 3) is fulfilled, as the maximum requirement is 32.5%.

As for the estimate of the amount of $CO_2$ (in kg) emitted into the atmosphere due to electric power generation, based on the energy mix at a national level, the emission factor is assumed to be equal to 283.6 g $CO_2$/kWh (ISPRA, 2015; http://www.isprambiente.gov.it/files/publication/rapporti/R_212_15.pdf (accessed on 27 May 2016)). Based on this assumption, and taking into account the $CO_2$ emission reductions by 2030 according

to paragraph 3.1.1, the $CO_2$ emissions should be reduced by 28%. Reconstruction of the buildings in the given locality will achieve a reduction in energy consumption of approximately 23.6 to 43.6%; so, when $CO_2$ production is reduced by 28%, the requirement of Directive 2012/27/EU is fulfilled as well.

Design of a Photovoltaic System (Photovoltaic Power Plant) on the Roofs of Selected Area of Buildings

The panel arrangement on the roofs of the building cluster is shown in Figure 18. Three options have been proposed:

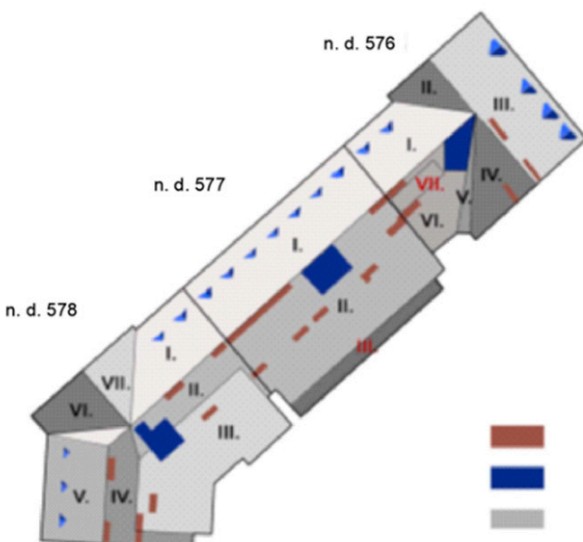

**Figure 18.** Demonstration of the roof surfaces of a cluster of buildings within an experiment.

Option 1–SOLAR-TERRA SOLRIF, monocrystalline silicon cells, output: 120 W/m$^2$, rated output: 90 Wp. This is a BIPV solar panel with the possibility of customized panel production. Gains from the installation under consideration: 44,992.3 kWh/year. The roof plane facing the courtyard with a total area of 470.4 m$^2$ was considered for installation.

Option 2–COPPO INVISIBLE SOLAR, monocrystalline silicon cells with a transparent layer, output: 83.3 W/m$^2$ (1 kWp/12 m$^2$). Rated output: 6 Wp. The gains from the installation under consideration on the area equal to 3/4 to 2/3 are 53,191.4 kWh/year.

Option 3–MAXEON 3400 W mono PERC, output: 226 W/m$^2$, rated output: 400 Wp. The gains from the installation under consideration on the area equal to 3/4 to 2/3 are 53,650.8 kWh/year.

Option 3, where the photovoltaic panel is installed on the roof of all the selected buildings, was the best one; it is not a panel with the possibility of integration into the roof plane. The suitability of this panel was also assessed in terms of aesthetic appearance. The installation is suitable for roofing facing the courtyard. These roof coverings are made up of a disparate material with no current requirement for appearance. It is a highly profitable option that will be effective even when placed on the part of the roof plane. Figure 18 shows the roof surfaces of the building cluster under our experiment.

For the next plant design, the least costly installation that achieves the highest profitability will be considered, namely Option 3 (MAXEON 3400 W).

### 3.4. Electric Power Consumption

The calculation of the locality's electricity consumption is based on a baseline that provides typical electric power consumption per m$^2$ of floor area for each type of building and each month of the year. The floor area was calculated from the floor plan of the individual buildings and the number of floors of the buildings. Figure 19 shows the electric power demand as a function of the building consumption in graphical form.

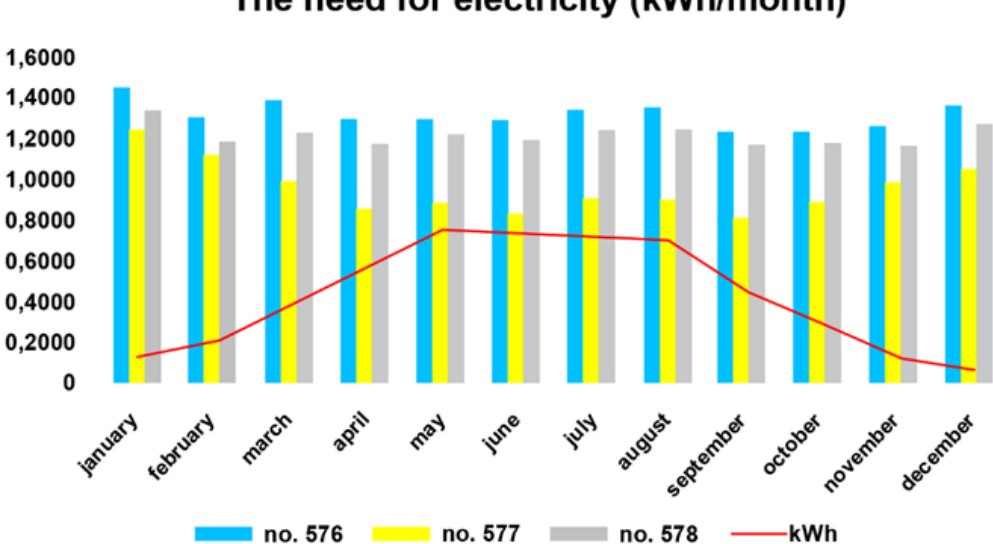

**Figure 19.** Electricity consumption by months of the year.

The graph shows that photovoltaics will only cover a small proportion of the energy needs. The design of other facilities will be in line with this finding, i.e., without the proposal to store surplus photovoltaic energy. Storage in the form of batteries would only be appropriate if they operated on a bulk remote control (BRC) signal and charged during low tariff to be used later in the day.

### 3.5. Cogeneration Unit Design

The cogeneration units will be located in the basement of building no. 578. The design is based on the annual heat demand for hot water. The cogeneration unit will also supply the electric power generated and will cover a part of the demand during its operation. TEDOM Cento 210 cogeneration unit is designed; thermal output: 241 kW, electric power: 210 kW, thermal efficiency: 46.5%, and electrical efficiency: 86.9%

### 3.6. Assessment Using Homer Pro

The MEC composition in its design form with Tesla Powerpack2 battery storage was considered for the assessment. It eventually proved unnecessary, given that the amount of energy generated by photovoltaics is consumed immediately without the need to store surpluses. Battery storage would only make sense if the battery could be charged during the low tariff period, and the energy from the battery could be used during the high tariff period. Figure 20 shows the simulation progress in the Homer software, The results section has two parts. The first, analytical scenarios, the second results of optimization. In the first part, we can select the scenario for which the results will be adjusted, otherwise the default values that were added in the wizard are taken into account. In the picture we see only one scenario with default values, the analysis tool was turned off during testing due to time dispositions.

In the next part we see how much energy and when we bought from the central network. We see that the energy bought from the grid in turn copies the production of electricity on the solar panel. The table also shows the energy purchased, the energy sold, the difference and the price in each of the months.

In Figure 21 we see only one scenario with default values. The analysis tool was turned off during testing due to time constraints. The results can be displayed graphically or in tabular form and can be exported to PDF.

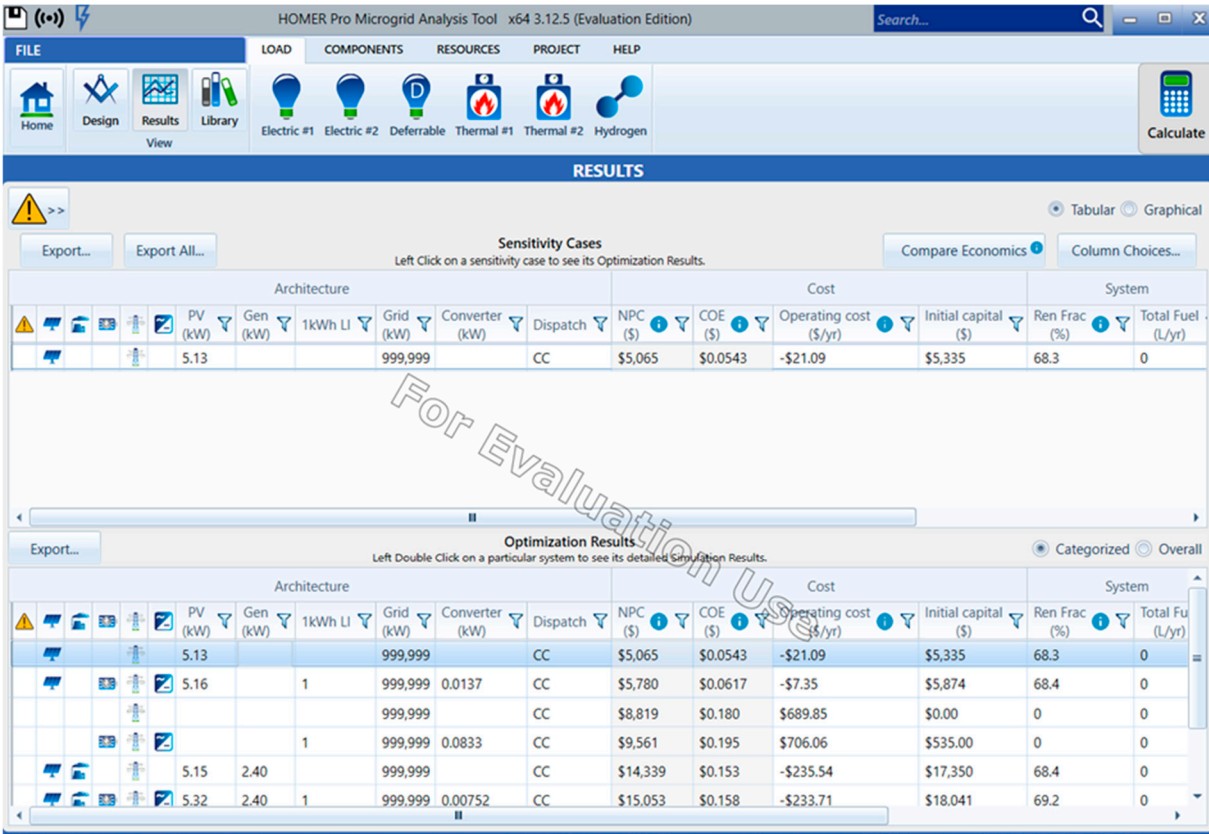

**Figure 20.** Výsledky simulace jedenoho scénáře s výchozími hodnotami.

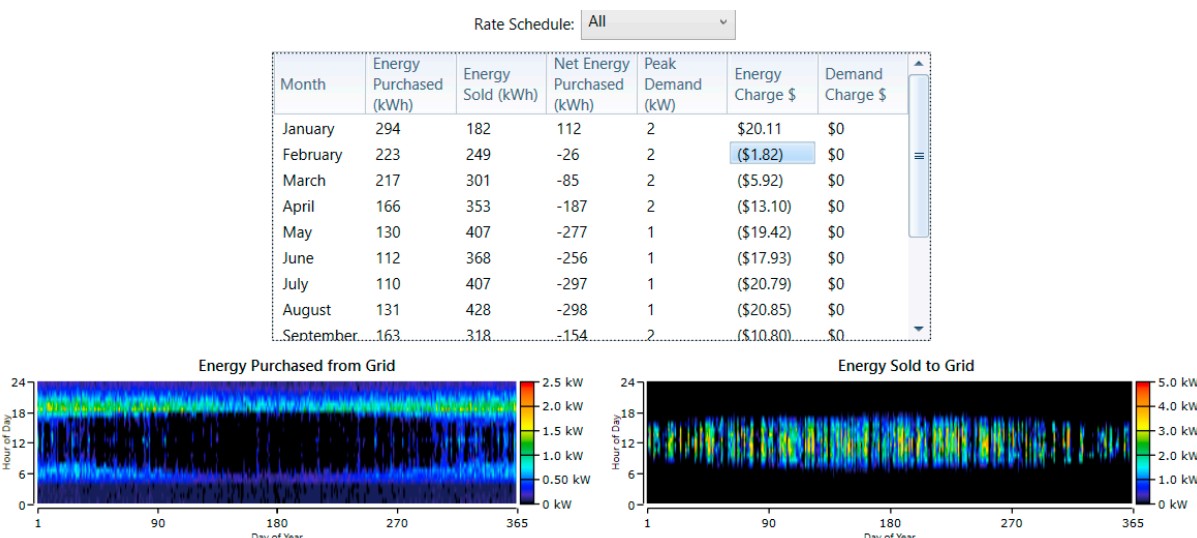

**Figure 21.** Simulations of produced and purchased energy and the table shows the price characteristics by months.

### 3.7. MEC Composition Design

Municipal energy centre for buildings nos. 576, 577, and 578 are located in the basement of building no. 577. It comprises: MAXON 3 photovoltaic panels placed on the roofs of buildings facing the courtyard, AC/DC inverters, and a TEDOM Cento 210 cogeneration unit. The switchboard will be located next to the existing electric power distribution board (in the basement of no. 578). Figure 15 shows a diagram of the MEC design for the given locality of our experiment.

## 4. Discussion

This work aims to solve the buildings' energy consumption and reduce energy consumption and $CO_2$ emission generation. The proposed measures are solved through EMB with IoT integration and the use of an automated control system, as well as KNX, FOXTRO, and ABB-free@home® control systems. The aim was also to ensure the energy sustainability of the defined area of the building cluster through a combination of building reconstruction and the use of a municipal energy centre (MEC). Given the local conditions, photovoltaic panels and a cogeneration unit were most appropriate to design. The use of integrated panels seemed to be the ideal solution in the initial design phase. Still, after a locality survey and comparison of options, a conventional option with photovoltaic panels was chosen.

When researching EPB solutions within a building or cluster of buildings, including municipal energy centres (MEC), three types of problems were found that need to be considered:

1. Approach to the solution–both the systemic and individual consequences of the solutions chosen need to be taken into account.
2. Analysis–parallel simulation helps to assess the benefits of different solutions for different selected systems.
3. Design, structure, and operation–choice of typology, new infrastructure capacity, and operations optimisation.

The analysis of the building cluster and the MEC highlights the possibility of simulating the behaviour of different parts of the system and the distribution of the output characteristics, namely the electric and thermal energy demand curves. In the context of a city, it is more efficient and convenient to use software with parallel simulation capabilities for analysis.

Such concurrent simulation, in most cases, points to different time scales of the buildings in question. It specifically notes the heat problems and considers the electric power infrastructure and relevant components under highly simplified hypotheses. Attempts have been made to simulate electrical and thermal systems simultaneously. In [61], a multidomain, customized simulation platform enables holistic analysis of building clusters and MEC. This platform allows long-term simulations of a large number of buildings, including internal energy supply or energy conversion systems together with exogenous energy sources such as the main power grid. Simulation of all these physical systems allows the evaluation of sophisticated energy management algorithms. However, the question of the usefulness of obtaining such a high level of detail in energy planning or management remains unanswered. For EPB cluster building modelling and MES, the literature discloses that other simulation tools are used quite a lot; however, these need to be evaluated in terms of specific sustainable energy requirements.

In the present case, the comprehensive EPB solution was based on the relevant decrees, standards, and laws, as commented in this article.

The current state of the EPB solution is defined, so the basic characteristics of NZEB buildings are fulfilled, which is supported by meeting the criteria given in Tables 1 and 2. In the case of a structurally designed solution according to Figure 6, the EPB is currently solved according to Figure 6, in steps 1 and 2. The output of the DesignBuilder simulation is the design of the building (building cluster) reconstruction in the urban area investigated, wherein the individual parameters according to the criteria of Tables 1 and 2 must be respected. The RES energy coverage of 30 kWh/m².y for a given house or building cluster was solved by the conventional method of designing photovoltaic systems (photovoltaic power plants). The indicators of the energy performance of a building comply with the current legislation, such as the total annual energy supplied, which is understood as the amount of energy supplied to the building and is expressed in kWh/m².year. In addition, the average heat transfer coefficient Uem was evaluated for certification purposes. In order to compare the permissible EPB values, the parameters determined by a reference building of the same type and set of characteristics listed in Decree 78/2013 Coll. were defined. As different software tools in the new EMB with IoT integration were used to simulate

parts of the concurrent simulation of each commodity, a wide range of software tools were analyzed from an energy planning perspective. None of these software tools deals with the issue of coupling individual characteristics of the partial energies themselves within the building, but this coupling needs to be considered in a city context where the city's layout significantly affects the consumption of all energies. The building simulation tool DesignBuilder; the RES microgrid optimisation and design software HOMER, including the PV*SOL software for photovoltaic system design and battery storage implementation; and Monte Carlo software for determining the electric power demand for electrical vehicles (EV) charging represent some of the most commonly used examples of these tools. Although in principle it should be possible to use these simulation tools for large-scale applications, there is no comprehensive study that reflects the modelling process, as well as the intercomparison of inputs, outputs, and the validity of these tools. For this reason, the future direction of our research has to be highlighted, which is a scientifically sound EPB modelling tool to compare inputs and outputs in terms of optimizing energy performance, energy sustainability, and energy efficiency.

## 5. Conclusions

Since 1 January 2013, an amendment to Act No. 406/2000 Coll. on energy management, as amended by Act No. 403/2020 Coll., has been in force in the Czech Republic, which has significantly changed and clarified the existing view on energy management. Measures to increase energy efficiency and other related obligations represent an important factor. Increased attention is to be focused on promoting energy savings, the use of renewable and secondary energy sources, and reducing emissions, particularly $CO_2$. This applies to all countries of the European Union (EU).

By applying RES, the assessment of the energy performance of a building and the final evaluation thereof in terms of the non-renewable energy factor, which is one of the main criteria for new and renovated buildings, can be influenced positively. It is no different concerning the operation and maintenance of buildings. The IoT technologies have been used in this sector for several years, most often in the field of environmental monitoring (various temperature, humidity, or $CO_2$ sensors), monitoring the status of building equipment (HVAC, heating, electric power consumption, etc.), or for remote energy readings.

IoE and IoT can be used to plan and address the transformation of European Union cities into smart and sustainable cities. Moreover, this transformation brings significant effects in the process of energy sustainability and environmental quality.

In this paper, the European and Czech legislation on Nearly Zero Energy Buildings (NZEBs) and smart cities was cited to show their importance when submitting technology proposals that are legally compatible and applicable to a smart city institution.

Figure 5 shows "Building energy model with the integration of IoT, Cloud and building control systems", and Figure 4 shows "Building energy model and KNX/FOXTROT control". Finally, the EPB was solved using the BEM, wherein the community energy was solved by IoT and IoE applications according to Figure 6, while the energy systems of technical building equipment are controlled by KNX, FOXTROT, or ABB-free@home® (indicated as "E" in Figure 6). The result of this complex process of EPB solutions according to BEM—see Figures 1 and 2—is a building (cluster of buildings) or a smart building (smart urban area) and thus a basic element of Smart Cities.

The process of addressing energy consumption optimisation is documented in Figure 10 (Energy consumption optimisation through IoT and building control systems). Subsequently, the IES was verified in the present experiment. For example, "smart" appliances can be remotely controlled and managed by a PLC (Programmable Logic Controller) or FOXTROT/KNX/ABB-free@home®, and information can be read from them. The design of the present solution for an automated IB control system based on IoT application is shown in Figure 13. In conjunction with the EPB solution aiming to reduce energy consumption and CO2 emissions significantly, a BEM was designed, where the proposed solution

(Figure 13) is implemented in the BEM (see Figure 5). Last but not least, the monitoring of energy consumption in existing buildings was solved. It was presented within the experiment by solving the building no. 578 (Figure 3). First of all, CSN 73 0540-2-2011 and the Commission Recommendation (EU) 2016/1318 of 29 July 2016 on guidelines for promoting nearly zero-energy buildings were referred to.

In addition, a comprehensive literature review is presented to gather all representative technological methods that can be applied for smart buildings, taking into account the European energy efficiency legislation. Then, a smart template for stable short- and long-term construction of energy-efficient buildings using IoT technology was presented. Prerequisites for designing a building management system that can contribute to the building certification and the control of building compliance with energy sustainability requirements and provide remote control and continuous measurement of all technical building systems have been created.

*Setting Up the EnergyHub Model in the GAMS Program Environment*

GAMS (General Algebraic Modeling System) is an algebraic modeling system suitable for mathematical optimization. This is an effective optimization method. We compared other optimization methods such as special optimization methods, including linear programming, dynamic programming, and convex programming, as well as another so-called combinatorial optimization. GAMS was the best for us.

We applied GAMS for the purpose of designing the city energy center (EnergyHub) in the defined area of Prague 6-Bubeneč. This is an effective optimization method that has been used for our purpose. We will briefly introduce the source codes for the compilation and calculation of the proposed variants (three variants) of the Energy Node.

To compile the program model code, we enter the input data, and then the GAMS program calculates the value of the purpose function. The basis for the calculation is the hourly energy consumption and electricity prices. These energy prices are always dependent on the current tariff. Figure 22 shows the necessary variables for time and the creation of a table of energy consumption, available power of photovoltaic panels, and hourly electricity prices).

```
1  $title Optimal operation of energy hub
2  */ creating the variable t
3  Set t 'hours' / t1*t24 /;
4  */ creating a data table with the variable t
5
6  Table data(t,*)
7  */ input values into the data table
8        Dh      De      Dg      PV       Lambda_e
9   t1   7844.38 112.50  45.00   0.00     2.13
10  t2   7844.38 97.50   45.00   0.00     2.13
11  t3   7844.38 90.00   45.00   0.00     2.13
12  t4   7844.38 90.00   45.00   0.00     2.13
13  t5   7844.38 97.50   45.00   0.00     2.13
14  t6   7844.38 112.50  45.00   0.00     2.13
15  t7   7844.38 142.50  45.00   0.00     2.13
16  t8   7844.38 172.50  45.00   0.00     2.13
17  t9   7844.38 187.50  45.00   212.89   2.13
18  t10  7844.38 187.50  45.00   456.28   2.13
19  t11  7844.38 195.00  45.00   608.17   2.55
20  t12  7844.38 210.00  45.00   626.22   2.55
21  t13  7844.38 202.50  45.00   412.39   2.55
22  t14  7844.38 180.00  45.00   181.79   2.13
23  t15  7844.38 195.00  45.00   7.24     2.13
24  t16  7844.38 210.00  45.00   0.00     2.13
25  t17  7844.38 255.00  45.00   0.00     2.13
26  t18  7844.38 307.50  45.00   0.00     2.13
27  t19  7844.38 315.00  45.00   0.00     2.13
28  t20  7844.38 307.50  45.00   0.00     2.13
29  t21  7844.38 285.00  45.00   0.00     2.55
30  t22  7844.38 247.50  45.00   0.00     2.55
31  t23  7844.38 195.00  45.00   0.00     2.55
32  t24  7844.38 150.00  45.00   0.00     2.13;
```

**Figure 22.** Input data code.

We will show only the module of the third EnergyHub program. Option 3 has components: transformer, cogeneration unit, gas boiler, battery storage, heat pump, and photovoltaic system. Variants 2 and 1 just mention the information of their composition. They contain the following components.

Option 2 has the following components: a transformer, a cogeneration unit, a gas boiler, a heat pump, and a photovoltaic system. Option 1 has the following components: transformer, cogeneration unit, gas boiler, and heat pump.

Based on the resulting value of the variable "cost", we find the most optimal solution in Figure 23.

```
33
34 */ introduction of the variable cost - operating costs
35 Variable cost;
36
```

**Figure 23.** Introduction of variables.

We will now declare variables for variant 3 Figure 24.

```
39 */ introduction of positive variables
40 Positive Variables E(t), E1(t), E2(t), E3(t), G(t), G1(t), G2(t), G3(t),
Ed(t), Ec(t),
41 H1(t), H_ehp(t), SOC(t) ;
42
43 */ introduction of binary variables
44 binary variables Ih(t), Idch(t), Ich(t);
45
46 $onEolCom
47 */ introduction of variables with given values
48 scalar eta_ee / 0.96 /,        !! efficient efficiency
49 eta_ge / 0.429 /,              !! efficiency of cogeneration unit production
electro
50 eta_gh / 0.456 /,             !! efficiency of cogeneration unit heat
production
51 eta_c / 0.9 /,                !! efficiency of battery storage charging
52 eta_d / 0.9 /,                !! efficiency of discharging the battery
storage
53 COP /4.1/,                    !! heat factor of the heat pump
54 H_ehpMax / 2000 / ,           !! heat pump output max
55 H_ehpMin / 0.5 / ,            !! heat pump output. min
56 Chpmax / 1400 / ,             !! max power of the cogeneration unit
57 Fmax / 6500 / ,               !! power gas boiler
58 eta_ghf / 0.915 / ,           !! efficiency of gas combustion by gas boiler
59 lambda_g / 1.231 / ,          !! price gas
60 SOCmax / 464 / ,              !! max battery storage charge status
61 SOC0 / 0 / ;                  !! min battery storage charge status
62
63 */ additional specification of variable values
64 H_ehp.up(t) = H_ehpMax ;      !! max heat pump output
65 G1.up(t)=Chpmax ;             !! max power CHP
66 G2.up(t)=Fmax ;               !! max power gas boiler
67 SOC0=0.2*SOCmax ;             !! initial state of bat. storage
68 SOC.up(t)=SOCmax ;            !! maximum battery capacity storage
69 SOC.lo(t)=0.2*SOCmax;         !! minimum capacity battery storage
70 SOC.fx('t24')=SOC0 ;          !! cycle bat. storage
71 EC.up(t)=0.2*SOCmax ;         !! maximum charging
72 EC.lo(t)=0 ;                  !! minimum charging
73 Ed.up(t)=0.2*SOCmax ;         !! maximum discharge
74 Ed.lo(t)=0 ;                  !! minimum discharge
75
```

**Figure 24.** Variables of the default mathematical model.

We will now introduce the equations into the GAMS program and run the solution to this problem, which is the following Figure 25.

Evaluation of results:

Option 1: The value of the objective function was evaluated at CZK 215,210.175. Compared to the original value of daily energy costs, we have a saving of 18.82%.

Option 2: The value of the objective function was evaluated at CZK 211,102.50. Compared to the original value of daily energy costs, we save 20.37%.

Option 3: The value of the objective function was evaluated at CZK 210,837.78. Compared to the original value of daily energy costs, we have savings of 20.47%.

The conclusion is that we will come up with the optimization of the purpose function, which we performed in the GAMS environment, and based on the results, we recommended to accept the composition of EnegyHub in variant 3.

The importance of the application of artificial intelligence in the process of energy savings and thus also the significant impact on the environment (reduction of $CO_2$ emissions) is clear. Currently selected methods of artificial intelligence allow us to solve optimization practical solutions. It was these applications that I used in our research. For example, the EH (EnergyHub) urban energy system was designed using the GAMS simulation software. Three EH variants were proposed and subsequently compared in terms of their energy efficiency and energy savings.

From the results of the partial solutions, we can see that the composition of EH in the system of photovoltaic panels and battery storage has led to a significant decrease in daily energy costs. In addition, these results were evaluated in a period of adverse weather conditions (autumn 2021).

Energy savings reached up to 20% of the original variant without optimization. If we researched these savings during the spring or summer, for example, we would achieve significantly better austerity measures. It is logical that the performance of individual EH components also depends on the environment. Significant energy savings directly affect emission reduction savings with similar characteristics, now up to 20% emission reductions.

```
75
76  */ introduction of the required number of equations with the notation
77  Equation eq1, eq2, eq3, eq4, eq5, eq6, eq7, eq8, eq9, eq10, eq11,
78  eq12, eq13, eq14, eq15, eq16 ;
79
80  */ equation defining the mathematical model EnergyHub:
81
82  */ objective evaluation function - sum of energy costs
83  eq1.. cost =e= sum(t, data(t,'lambda_e')*E(t)+lambda_g*G(t)):
84  */ power flow from EnergyHub
85  eq2(t).. E2(t)+eta_ge*G1(t)+Ed(t) =e= data(t, 'De')+E3(t);
86  */ power flow to the EnergyHub
87  eq3(t).. eta_ee*E(t) + data(t, 'PV') =e= E1(t) + E2(t) ;
88  */ defining the input of electrical energy into the battery storage
89  eq4(t).. E1(t) =e= Ec(t) ;
90  */ change the state of the battery storage charge level
91  eq5(t).. SOC(t) =e= SOC0$(ord(t)=1)+SOC(t-1)$(ord(t)> 1)+Ec(t)*eta_c-
Ed(t)/eta_d ;
92  */ battery storage discharge limit
93  eq6(t).. Ed(t) =l= 0.2*SOCmax*Idch(t) ;
94  */ battery storage charging limit
95  eq7(t).. Ec(t) =l= 0.2*SOCmax*Idch(t) ;
96  */ battery storage operating mode (charging / discharging)
97  eq8(t).. Idch(t)+Ich(t) =l= 1 ;
98  */ gas flow to EnergyHub
99  eq9(t).. G(t) =e= G1(t)+G2(t)+G3(t) ;
100 */ gas flow from EnergyHub
101 eq10(t).. G3(t) =e= data(t, 'Dg') ;
102 */ heat flow from EnergyHub
103 eq11(t).. eta_gh*G1(t)+H1(t)+H_ehp(t) =e= data(t, 'Dh') ;
104 */ gas boiler operation
105 eq12(t).. eta_ghf*G2(t) =e= H1(t) ;
106 */ heat pump output
107 eq13(t).. H_ehp(t) =e= E3(t)*COP ;
108 */ limitation of the maximum heat output of the heat pump
109 eq14(t).. H_ehp(t) =l= H_ehpMax*Ih(t) ;
110 */ limitation of the minimum heat output of the heat pump
111 eq15(t).. H_ehp(t) =g= H_ehpMax*Ih(t)*H_ehpMin ;
112 */ heat pump operating mode (cooling / heating)
113 eq16(t).. Ih(t) =l= 1 ;
114
115 */ creating a mathematical model from the entered data
116 Model Hub / all /;
117
118 */ command for solving the given model using MIP solver with
minimization of the variable cost
119 solve hub us mip min cost ;
120
```

**Figure 25.** Formulation of mathematical model equations in GAMS code.

**Funding:** This research received no external funding.

**Institutional Review Board Statement:** Not applicable.

**Informed Consent Statement:** Not applicable.

**Data Availability Statement:** The data presented in this study are available upon request from the respective author.

**Conflicts of Interest:** The author declares no conflict of interest.

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
