# Peer review of "Energy Centers in a Smart City as a Platform for the Application of Artificial Intelligence and the Internet of Things"

_applsci, doi:10.3390/app12073386_

Round 1

Reviewer 1 Report

The research on the process management of sustainable energy in smart cities is meaningful. However, I don’t find many contributions and the novelty in this paper, and some indistinct expressions abate the readability of this paper simultaneously. Some comments are listed below:

1. This paper is more like a work report, without innovative research methods and optimization methods.

2. The framework in Fig. 2 is an ordinary one and is common in other studies. Authors should highlight their major contributions in the context.

3. More detailed comments on the colormap in the paper would helpful.

4. It is recommended to add effective optimization methods based on the conclusions.

Reviewer 2 Report

The main objective of the article is to design a smart building that serves as a model to be replicated within a smart city. The article presents a use case of a building that monitors the performance of a range of devices through IoT technology in order to achieve energy efficiency and sustainable energy within a smart city.

The article is relevant within the field of energy efficiency with IoT devices, however in my opinion it is not very well structured and should be restructured. The experimental design is adequate to test the hypothesis, but the order makes it difficult to interpret it correctly. And finally the conclusions are coherent with the evidence and the arguments presented.

Below I list some details that I have found in the manuscript that can be improved:

  • in line 458 put a bullet
  • line 467 indentation
  • 517 MOM is not a software it's a paradigm
  • 518 it's obvious, that's what a protocol does
  • concepts such as "SCADA", "Thread" suddenly appear without proper introduction
  • use the full name of "EMB" on line 557
  • From line 673 the text is in italics, any reason?
  • The paragraph from line 692 is too general or you can merge it with the part 2.1 or make it specific for the example
  • The whole paragraph of active energy line 768 is general and becomes even obvious and that title has nothing to do with cloud computing.
  • The title of the article talks about artificial intelligence but in the text there is little or nothing about algorithms or models for energy or environmental savings.

Author Response

Reviewer No. 2

Reviewer's statement and response to this statement

When it comes to the structure of the article, I tried to make it chronological. Although I have no major comments against your statement. Its restructuring would require a comprehensive reorganization of the text, and given the original structure proposal, I would "přehupl" undermine the MDPI's recommendations on how to structure the article.

Nevertheless, thank you very much and for the next time I will definitely pay close attention to the structure of scientific articles !!!

Re point 1:

I put a dot in line 458.

Re point 2:

I indented line 467.

Re point 3:

I inserted the sentence on line 542:

Message oriented middleware (MOM) is a type of software product that enables message distribution over complex IT systems.

Re point 4:

Line 543. I agree with your statement, I also think so.

Re point 5:

I have included in the article chronologically article 2.1 SCADA services. This is why I explained this concept.

Re point 6:

I changed the name of EMB in the text to the correct name (building energy model) BEM, which is explained on page 9. This is how it is used throughout the article.

Re point 7:

I have changed all italics in the text to the classic form. It didn't matter. I just want to distinguish it visually, as a definition or example.

Re point 8:

I have excluded the general paragraph in lines 717 to 727 from the text. It was a general example. You're right!

Re point 9:

I excluded from the text the whole paragraph from lines 802 to 823, ie the whole paragraph called Active Energy. I agree, you're right. I am sorry.

Re point 10:

Note: The title of the article.

The article talks about artificial intelligence. I included optimization models, GAMS applications for solving the optimization of the city energy center (EnergyHub). The GAMS application was performed within the experiment and is inserted in the text at the end of the article. When solving the new design of the energy center, we will achieve significant energy savings and thus reduce CO2. At the end of the paper, the applications of artificial intelligence methods were evaluated.

Reviewer 3 Report

Dear Authors,

The paper is well written and has a good contribution to the expertise. The revisions are made well.

Author Response

Reviewer No. 3

There are no comments for reviewer # 3. He agrees with the article.

Reviewer 4 Report

The author has presented an interesting paper on an important problem involving electricity demand in smart cities. The article is very interesting, however there are a few remarks that need improvement:
- the article is too long, which despite interesting research results may discourage the reader from a thorough analysis
- there are some editorial errors - different font in the section titles or shifting of the tables (e.g. Table 1), please refine the text in this regard 
- non-English units appear in Formulas 26, 27

Author Response

Reviewer No. 4

Re point 1:

A reminder that the article is long. I agree. But on the other hand, I must note that the topic is very challenging, current, and therefore requires some explanation of new concepts and new solutions.

Re point 2:

I have modified Table 1 to the title: Reference values for EPUB and NZEB for countries in different EU climate zones. I also adjusted the font in the section names according to the template.

Re point 3:

I corrected the English unit in relations (26) and (27).